# Revisiting Convergence of AdaGrad with Relaxed Assumptions

**Yusu Hong**[1]  **Junhong Lin**[1]

[1]Zhejiang University

## Abstract

In this study, we revisit the convergence of Ada-Grad with momentum (covering AdaGrad as a special case) on non-convex smooth optimization problems. We consider a general noise model where the noise magnitude is controlled by the function value gap together with the gradient magnitude. This model encompasses a broad range of noises including bounded noise, sub-Gaussian noise, affine variance noise and the expected smoothness, and it has been shown to be more realistic in many practical applications. Our analysis yields a probabilistic convergence rate which, under the general noise, could reach at $\tilde{\mathcal{O}}(1/\sqrt{T})$. This rate does not rely on prior knowledge of problem-parameters and could accelerate to $\tilde{\mathcal{O}}(1/T)$ where $T$ denotes the total number iterations, when the noise parameters related to the function value gap and noise level are sufficiently small. The convergence rate thus matches the lower rate for stochastic first-order methods over non-convex smooth landscape up to logarithm terms [Arjevani et al., 2023]. We further derive a convergence bound for AdaGrad with momentum, considering the generalized smoothness where the local smoothness is controlled by a first-order function of the gradient norm.

## 1 INTRODUCTION

In recent years, AdaGrad [Duchi et al., 2011] and its variants have witnessed a large success in solving the following stochastic optimization problems:

$$\min_{\boldsymbol{x} \in \mathbb{R}^d} f(\boldsymbol{x}), \quad \text{where} \quad f(\boldsymbol{x}) = \mathbb{E}_{\boldsymbol{\zeta}}[f_{\boldsymbol{\zeta}}(\boldsymbol{x}; \boldsymbol{\zeta})].$$

Distinct from vanilla Stochastic Gradient Descent (SGD) [Robbins and Monro, 1951], which typically requires smoothness or Lipschitz constants for tuning step-sizes, AdaGrad applies an adaptive step-size with each coordinate satisfying that

$$\eta_{t,i} = \frac{\eta}{\sqrt{\sum_{s=1}^{t} g_{s,i}^2 + \epsilon}}, \quad \forall t \in \mathbb{N}, i \in [d],$$

where $\epsilon > 0$ is a constant and $g_{s,i}$ denotes the $i$-th coordinate of the stochastic gradient $\boldsymbol{g}_s$. This approach assigns larger step-sizes for infrequent features whose corresponding gradients are small, reminding learners of taking notice of those infrequent features. It also liberates the algorithm from the need for problem-parameters, which may be challenging to obtain in practical applications. Moreover, Ada-Grad's efficiency has been empirically validated, especially in scenarios with sparse gradients [Duchi et al., 2011].

Numerous works have studied the convergence of AdaGrad and its scalar version, AdaGrad-Norm [Duchi et al., 2011, Streeter and McMahan, 2010]. Duchi et al. [2011] first provided the convergence bound of AdaGrad on online convex optimization. In non-convex smooth scenario, Ward et al. [2020] first obtained a convergence bound for AdaGrad-Norm without pre-tuning step-sizes, assuming bounded gradients and noises. Liu et al. [2023b] proved the convergence for AdaGrad under coordinate-wise sub-Gaussian noise, discarding the bounded gradient assumption.

Recently, several studies have proven AdaGrad-Norm's convergence under the affine variance noise, both in expectation [Faw et al., 2022, Wang et al., 2023] and in high probability [Attia and Koren, 2023]. The noise model assumes that the stochastic gradient $g(\boldsymbol{x}), \forall \boldsymbol{x} \in \mathbb{R}^d$ satisfies that for some constants $B, C > 0$,

$$\begin{aligned} \mathbb{E}\|g(\boldsymbol{x}) - \nabla f(\boldsymbol{x})\|^2 &\leq B\|\nabla f(\boldsymbol{x})\|^2 + C \\ \text{or} \quad \|g(\boldsymbol{x}) - \nabla f(\boldsymbol{x})\|^2 &\leq B\|\nabla f(\boldsymbol{x})\|^2 + C. \end{aligned} \tag{1}$$

This noise model, verified in machine learning applications with feature noise [Fuller, 2009, Khani and Liang, 2020], and in robust linear regression [Xu et al., 2008], offers a

*Accepted for the 40$^{th}$ Conference on Uncertainty in Artificial Intelligence* (UAI 2024).

more realistic portrayal by allowing the noise norm to increase with the gradient norm, covering both bounded noise and sub-Gaussian noise. These studies not only provided a convergence rate of $\tilde{\mathcal{O}}(1/\sqrt{T})$, but also addressed challenges posed by the entanglement of adaptive step-sizes and stochastic gradients, and the additional variance in (1). However, to the best of our knowledge, none of existing works have proved the convergence of vanilla AdaGrad under (1) without assuming bounded gradients. Moreover, the distinct step-size for each coordinate in AdaGrad, as opposed to an unified step-size for all coordinates in AdaGrad-Norm, brings more challenges when considering (1).

In this paper, we provide a deep analysis framework and establish a probabilistic convergence bound for AdaGrad with heavy-ball style momentum, covering AdaGrad as a special case. More importantly, we consider a general noise model such that for some constants $A, B, C \geq 0$,

$$\|g(\boldsymbol{x}) - \nabla f(\boldsymbol{x})\|^2 \leq A(f(\boldsymbol{x}) - f^*) + B\|\nabla f(\boldsymbol{x})\|^2 + C,$$

$$(2)$$

where $f(\boldsymbol{x}) \geq f^*, \forall \boldsymbol{x} \in \mathbb{R}^d$. It's obvious to verify that (2) is strictly weaker than the almost surely affine variance noise in (1), and thus than the bounded noise or sub-Gaussian noise condition[1]. Indeed, (2) could be regarded as an extension of (1) and the following expected smoothness condition [Gower et al., 2019, Grimmer, 2019, Wang and Yuan, 2023],

$$\mathbb{E}\|g(\boldsymbol{x})\|^2 \leq A(f(\boldsymbol{x}) - f^*) + B$$
$$\text{or} \quad \|g(\boldsymbol{x})\|^2 \leq A(f(\boldsymbol{x}) - f^*) + B. \quad (3)$$

Existing researches have studied SGD's convergence behavior under (2) with smooth objective functions, both in asymptotic [Poljak and Tsypkin, 1973] and non-asymptotic view [Khaled and Richtárik, 2023]. More importantly, it has been shown that numerous of practical stochastic gradient settings satisfy (2) but out of the range of (1), including commonly used perturbation, sub-sampling and compression [Khaled and Richtárik, 2023]. However, the analysis for SGD could not be directly extended to AdaGrad due to the correlation of adaptive-sizes and stochastic gradients, and the coordinate-wise performance in AdaGrad.

Finally, we apply our analysis framework to the $(L_0, L_1)$-smoothness where the local smoothness of $f$ satisfies that when $\|\boldsymbol{y} - \boldsymbol{x}\| \leq 1/L_1$,

$$\|\nabla f(\boldsymbol{y}) - \nabla f(\boldsymbol{x})\| \leq (L_0 + L_1\|\nabla f(\boldsymbol{x})\|)\|\boldsymbol{y} - \boldsymbol{x}\|. \quad (4)$$

This assumption was proposed by [Zhang et al., 2020b] through empirical studies on language models and later verified in large language models, e.g., [Zhang et al., 2020a,

Crawshaw et al., 2022]. (4) generalizes the standard global smoothness condition and allows unbounded smooth parameter, bringing more challenges for the convergence analysis of adaptive methods. Previous works [Faw et al., 2023, Wang et al., 2023] have derived convergence bounds for AdaGrad-Norm with (1) and (4). Also, prior knowledge of problem-parameters is necessary as pointed out by the counter examples in [Wang et al., 2023]. However, the analysis for coordinate-wise AdaGrad is non-trivial and requires more delicate constructions, particularly when considering the weaker noise assumption in (2).

In the following, we will summarize our main contributions as follows. We also refer readers to see the comparison of our results with existing works in Table 1 from the appendix.

**Contribution**

- We demonstrate the probabilistic convergence of AdaGrad with momentum on non-convex smooth optimization under a general noise assumption in (2). For an $L$-smooth function $f$, we demonstrate that after $T$ iterations of the algorithm, with probability at least $1 - \delta$, $\sum_{s=1}^{T} \|\nabla f(\boldsymbol{x}_s)\|^2 / T$ is bounded by

$$\mathcal{O}\left(\frac{\text{poly}\left(\log \frac{T}{\delta}\right)}{T} + \sqrt{\frac{(A+C)\text{poly}\left(\log \frac{T}{\delta}\right)}{T}}\right),$$

which could also accelerate to $\tilde{\mathcal{O}}(1/T)$ rate when the noise parameters $A, C$ are sufficiently low.

- As direct corollaries, we also derive similar probabilistic convergence results of AdaGrad on non-convex smooth optimization with affine variance noise. More importantly, the convergence rate is optimal and adaptive to the noise level $C$ in (1).

- We derive a convergence bound for AdaGrad with momentum considering (2) and (4). The rate is similar to the smooth case and adaptive to the noise level as well, and necessitating problem-parameters to tune step-sizes.

Our analysis relies on the descent lemma with telescoping, and the novel decomposition and estimations over the first-order term related to new proxy step-sizes that are used to decorrelate stochastic gradients and adaptive step-sizes in this new noise regime. We also prove that the function value gap as well as the gradient norm are controlled by the polynomial of $\log T$ along the optimization process.

The rest of the paper are organized as follows. The next section introduces some extra related works. Section 3 provides the problem setup and basic assumptions, and the introduction of AdaGrad with momentum. Section 4 provides high probability convergence bounds for AdaGrad with momentum, and also for AdaGrad as direct corollaries. Section 5 provides proof details for the main results. Section

---

[1]For conciseness, we mainly consider the almost-sure version of this general noise model. Extending our high probability analysis from an almost-sure version to a sub-Gaussian version is easy, which will be included in Appendix.

6 presents necessary introduction of the generalized smooth condition and the subsequent convergence result. All missing proofs for some of the lemmas and convergence results under generalized smoothness are given in Appendix.

## 2 RELATED WORKS

SGD and its adaptive variants have been a target of intense interest in the last decade. We refer to [Bottou et al., 2018, Ruder, 2016] for an overview. We limit our discussions to the most relevant literature in the sequel.

**Convergence of AdaGrad**  Numerous of works mainly studied the convergence of AdaGrad-Norm over non-convex smooth landscape. Li and Orabona [2019] first proved the convergence for AdaGrad-Norm. However, they studied a variant with a delayed step-size that is independent from the current stochastic gradient and required knowledge of the smoothness parameter for tuning step-sizes. Getting rid of prior-knowledge on problem-parameters, Ward et al. [2020] relied on a novel proxy step-size technique and showed the convergence with an uniform bound of stochastic gradients for vanilla AdaGrad-Norm. Kavis et al. [2022] and Liu et al. [2023b] proved probabilistic convergence under the sub-Gaussian noise without relying on the bounded gradient assumption. In case of the affine variance noise, Faw et al. [2022] provided the convergence bound. However, their rate is adaptive to the noise level only when $B \sim \mathcal{O}(1/T)$. Wang et al. [2023] relied on a distinct framework to improve the dependency on $T$ in the convergece rate [Faw et al., 2022] and achieved the adaptivity on noise level without any restriction over $B$. Concurrently, Attia and Koren [2023] deduced a probabilistic bound, using a novel induction argument to control the function value gap. Their result also adapted to the noise level without further requirement on $B$. Liu et al. [2023a] formulated a convergence bound for AdaGrad-Norm and its acceleration version in quasar-convex smooth setting.

The element-wise version of AdaGrad was first studied in [Duchi et al., 2011] on online convex optimization. In non-convex smooth case, a line of works investigated AdaGrad with bounded gradients. Zou et al. [2019] explored the convergence of AdaGrad with a heavy-ball or Nesterov style momentum. Zhou et al. [2020] also covered AdaGrad in their analysis, but they deduced a bound under requiring bounded gradients' summation. Défossez et al. [2022] studied AdaGrad with Adam-type momentum and improved the dependency on the momentum parameter $\beta$ to $\mathcal{O}((1-\beta)^{-1})$. Shen et al. [2023] introduced a weighted AdaGrad with unified momentum covering both heavy-ball and Nesterov's acceleration. Recently, Liu et al. [2023b], a work mentioned before, derived a convergence bound under coordinate-wise sub-Gaussian noise, i.e., $g(\boldsymbol{x})_i - \nabla f(\boldsymbol{x})_i$ is sub-Gaussian for each $i \in [d]$, without requiring bounded gradients.

**Convergence with affine variance noise**  We briefly summarize some works on the convergence of SGD or AdaGrad with (1) and its variants under the non-convex smooth landscape. Bertsekas and Tsitsiklis [2000] provided an almost-surely convergence bound for SGD. In non-asymptotic view, Bottou et al. [2018] derived a convergence bound for SGD of the form $\mathcal{O}(1/T + \sqrt{C/T})$ when step-sizes are well tuned by the smooth parameter, $B$ and $C$. They also pointed out that the extension is immediate from the bounded noise case [Ghadimi and Lan, 2013]. The convergence of AdaGrad-Norm under (1) has been well studied by [Faw et al., 2022, Wang et al., 2023, Attia and Koren, 2023] as mentioned before. Faw et al. [2023] further extended the analysis considering a generalized smooth condition. However, none of these existing works could prove the convergence of coordinate-wise version of AdaGrad under (1) or a weaker noise condition.

**Convergence with the expected smoothness**  The expected smoothness condition was once applied for convex optimization such that for some constant $A > 0$,

$$\mathbb{E}[\|g(\boldsymbol{x}) - g(\boldsymbol{x}^*)\|^2] \leq A(f(\boldsymbol{x}) - f^*), \qquad (5)$$

where $\boldsymbol{x}^*$ denotes the global minimizer. Based on (5), Richtárik and Takác [2020] relied on matrix analysis to bound the identities of expected iterates of SGD in the setting of stochastic reformulations of linear systems. Gower et al. [2021] applied (5) to analyze the JacSketch method (a general form of SAGA) over strongly convex optimization.

Since $\boldsymbol{x}^*$ is ill-defined for non-convex optimization, Gower et al. [2019] then directly set $\mathbb{E}[\|g(\boldsymbol{x}^*)\|^2] = B$ and deduced the non-convex version of the expected smoothness in (3), which aligns with the weak growth condition [Vaswani et al., 2019] when $B = 0$. Gower et al. [2019] relied on (3) to analyze SGD over quasi-strongly convex optimization. Independently, Grimmer [2019] relied on (3) and developed a general framework for SGD equipped with projection operators over convex non-smooth functions. Wang and Yuan [2023] also used (3) to derive a convergence bound for SGD using bandwidth-based step-sizes.

Regarding the noise model in (2), Poljak and Tsypkin [1973] provided an asymptotic convergence bound for SGD with smooth objective functions. Very recently, Khaled and Richtárik [2023] derived a non-asymptotic convergence rate of $\mathcal{O}(1/\sqrt{T})$ for SGD with non-convex smooth functions.

In conclusion, it's clear to see that (2) is weaker than the above conditions including (1) (when assuming the existence of $f^*$), (3) and (5)[2]. Our result then shows that AdaGrad could find a stationary point under this mild noise assumption without prior knowledge of problem-parameters.

---

[2]We make the comparison when assuming all conditions are in almost-surely form.

**Convergence with generalized smoothness** The generalized smooth condition [Zhang et al., 2020b] has been well studied under different algorithms, e.g., [Qian et al., 2021, Zhao et al., 2021, Reisizadeh et al., 2023, Zhang et al., 2020a, Crawshaw et al., 2022]. Considering AdaGrad and its variants, Faw et al. [2023] established a convergence bound for AdaGrad-Norm considering (1). However, their result required $B < 1$. Wang et al. [2023] further tightened the dependency to the iteration number $T$ and got rid of restriction on $B$.

# 3 PROBLEM SETTING AND ALGORITHM

We consider unconstrained stochastic optimization over the Euclidean space $\mathbb{R}^d$ with $l_2$-norm. The objective function $f : \mathbb{R}^d \to \mathbb{R}$ is $L$-smooth satisfying that for any $\boldsymbol{x}, \boldsymbol{y} \in \mathbb{R}^d$,

$$f(\boldsymbol{y}) - f(\boldsymbol{x}) - \langle \nabla f(\boldsymbol{x}), \boldsymbol{y} - \boldsymbol{x} \rangle \le \frac{L}{2} \|\boldsymbol{x} - \boldsymbol{y}\|^2.$$

Given $\boldsymbol{x} \in \mathbb{R}^d$, we assume a gradient oracle that returns a random vector $g(\boldsymbol{x}, \boldsymbol{z}) \in \mathbb{R}^d$, where $\boldsymbol{z}$ denotes a random sample. The deterministic gradient of $f$ at $\boldsymbol{x}$ is denoted by $\nabla f(\boldsymbol{x}) \in \mathbb{R}^d$.

**Notations** We denote the set $\{1, 2, \cdots, T\}$ as $[T]$, and use $\|\cdot\|, \|\cdot\|_1$, and $\|\cdot\|_\infty$ to represent the $l_2$-norm, $l_1$-norm, and $l_\infty$-norm, respectively. The notations $a \sim \mathcal{O}(b)$ and $a \le \mathcal{O}(b)$ refer to $a = c_1 b$ and $a \le c_2 b$ with $c_1, c_2$ being positive universal constants, and $a \le \tilde{\mathcal{O}}(b)$ indicates $a \le \mathcal{O}(b)\text{poly}(\log b)$. For any vector $\boldsymbol{x} \in \mathbb{R}^d$, the expressions $\boldsymbol{x}^2$ and $\sqrt{\boldsymbol{x}}$ refer to the coordinate-wise square and square root. For two vectors $\boldsymbol{x}, \boldsymbol{y} \in \mathbb{R}^d$, $\boldsymbol{x} \odot \boldsymbol{y}$ and $\boldsymbol{x}/\boldsymbol{y}$ denote the coordinate-wise product and quotient. $\boldsymbol{0}_d$ and $\boldsymbol{1}_d$ signify zero and one vectors in $d$ dimensions. Further, we write $\boldsymbol{1}_d/\boldsymbol{x}$ as $1/\boldsymbol{x}$, whenever there is no any confusion.

**Assumption** We make the following assumptions.

- **(A1) Bounded below:** The objective function is bounded below, i.e., there exists $f^* > -\infty$ such that $f(\boldsymbol{x}) \ge f^*, \forall \boldsymbol{x} \in \mathbb{R}^d$;

- **(A2) Unbiased estimator:** The gradient oracle provides an unbiased estimator of $\nabla f(\boldsymbol{x})$, i.e., $\forall \boldsymbol{x} \in \mathbb{R}^d$, $\mathbb{E}_{\boldsymbol{z}}[g(\boldsymbol{x}, \boldsymbol{z})] = \nabla f(\boldsymbol{x})$;

- **(A3) Relaxed affine variance noise:** The gradient oracle satisfies that for some constants $A, B, C > 0$, $\|g(\boldsymbol{x}, \boldsymbol{z}) - \nabla f(\boldsymbol{x})\|^2 \le A(f(\boldsymbol{x}) - f^*) + B\|\nabla f(\boldsymbol{x})\|^2 + C, a.s., \forall \boldsymbol{x} \in \mathbb{R}^d$.

The first two assumptions are standard in the analysis of algorithm's convergence. With a simple calculation, it's easy to verify that Assumption (A3) is equivalent to

$$\|g(\boldsymbol{x}, \boldsymbol{z})\|^2 \le A'(f(\boldsymbol{x}) - f^*) + B'\|\nabla f(\boldsymbol{x})\|^2 + C'$$

for another three positive constants $A', B', C'$. Therefore, (A3) is a generalization of (1) and (3). For more detailed examples of stochastic gradient settings satisfying Assumption (A3), we refer interested readers to see [Khaled and Richtárik, 2023, Proposition 2,3].

---

**Algorithm 1** AdaGrad with momentum

---

**Input:** Horizon $T$, $\boldsymbol{x}_1 \in \mathbb{R}^d$, $\beta \in [0, 1)$, $\boldsymbol{m}_0 = \boldsymbol{v}_0 = \boldsymbol{0}_d$, $\eta, \epsilon > 0$, $\boldsymbol{\epsilon} = \epsilon \boldsymbol{1}_d$
  **for** $s = 1, \cdots, T$ **do**
    Draw a random sample $\boldsymbol{z}_s$ and generate $\boldsymbol{g}_s = g(\boldsymbol{x}_s, \boldsymbol{z}_s)$;
    $\boldsymbol{v}_s = \boldsymbol{v}_{s-1} + \boldsymbol{g}_s^2$;
    $\boldsymbol{m}_s = \beta \boldsymbol{m}_{s-1} - \eta \boldsymbol{g}_s / (\sqrt{\boldsymbol{v}_s} + \boldsymbol{\epsilon})$;
    $\boldsymbol{x}_{s+1} = \boldsymbol{x}_s + \boldsymbol{m}_s$;
  **end for**

---

**AdaGrad with momentum** Throughout the paper, we study AdaGrad with momentum given in Algorithm 1. We can transform Algorithm 1 into the classical Polyak's heavy-ball method [Polyak, 1964] with an adaptive step-size:

$$\boldsymbol{x}_{s+1} = \boldsymbol{x}_s - \eta \frac{\boldsymbol{g}_s}{\sqrt{\boldsymbol{v}_s} + \boldsymbol{\epsilon}} + \beta(\boldsymbol{x}_s - \boldsymbol{x}_{s-1}), \ \forall s \in [T], \quad (6)$$

where we set $\boldsymbol{x}_0 = \boldsymbol{x}_1$.

**AdaGrad** AdaGrad is Algorithm 1 with $\beta = 0$.

# 4 MAIN CONVERGENCE RESULT

In this section, we provide the probabilistic convergence result for Algorithm 1 under Assumption (A3) and smooth objective functions.

**Theorem 1.** *Given $T \ge 1$, let $\{\boldsymbol{x}_s\}_{s \in [T]}$ be generated by Algorithm 1. If Assumptions (A1), (A2), (A3) hold, then for any $\beta \in [0, 1), \eta, \epsilon > 0$ and $\delta \in (0, 1)$, it holds that with probability at least $1 - \delta$,*

$$\frac{1}{T} \sum_{s=1}^{T} \|\nabla f(\boldsymbol{x}_s)\|^2$$
$$\le \mathcal{O}\left[ \Delta_1 \left( \frac{B_1 \Delta_1 + \sqrt{B_1 L \Delta} + \epsilon}{T} + \sqrt{\frac{A\Delta + C}{T}} \right) \right],$$

*where $B_1 = B + 1$, $\Delta_1 = \Delta(1 - \beta)/\eta$, and $\Delta$ is given by*[3]

$$\Delta \sim \mathcal{O}\left[ f(\boldsymbol{x}_1) - f^* + \frac{\sqrt{C}\eta d}{1 - \beta} \log\left( \frac{T}{\delta} + \frac{T}{\epsilon^2} \right) \right.$$
$$\left. + \frac{(A + B_1 L)\eta^2 d^2}{(1 - \beta)^3} \log^2\left( \frac{T}{\delta} + \frac{T}{\epsilon^2} \right) \right].$$

---

[3]The detailed expression of $\Delta$ could be found in (20).

**Remark 4.1.** *With a simple calculation, when $\eta = c_1(1 - \beta)^{3/2}$ for some constant $c_1 > 0$, the above upper bound has a minimum order of $\mathcal{O}((1-\beta)^{-1})$ with respect to $(1-\beta)^{-1}$. The comparison of existing results with our convergence bound could be found in Table 1 from the appendix.*

**Convergence of AdaGrad with affine variance noise** As a direct consequence of Theorem 1, it's worthy to mention the following convergence bound for AdaGrad with affine variance noise considering their empirical significance.

**Corollary 1.** *Under the assumptions and notations of Theorem 1, let $\beta = 0$ and $A = 0$. Then for any $\eta, \epsilon > 0$ and $\delta \in (0, 1)$, it holds that with probability at least $1 - \delta$,*

$$\frac{1}{T} \sum_{s=1}^{T} \|\nabla f(\boldsymbol{x}_s)\|^2$$
$$\leq \mathcal{O}\left[\Delta_1 \left(\frac{B_1\Delta_1 + \sqrt{B_1 L\Delta} + \epsilon}{T} + \sqrt{\frac{C}{T}}\right)\right],$$

*where $B_1 = B + 1$, $\Delta_1 = \Delta/\eta$, and $\Delta$ is defined as follows*

$$\Delta \sim \mathcal{O}\left[f(\boldsymbol{x}_1) - f^* + \sqrt{C}\eta d \log\left(\frac{T}{\delta} + \frac{T}{\epsilon^2}\right)\right.$$
$$\left. + B_1 L\eta^2 d^2 \log^2\left(\frac{T}{\delta} + \frac{T}{\epsilon^2}\right)\right].$$

**Remark 4.2.** *1) Setting $\eta \sim 1/\left(d \log\left(\frac{T}{\delta}\right)\right)$, then $\Delta \sim 1$, and the above derived upper bound is of order $\mathcal{O}\left(d^2 \log^2\left(\frac{T}{\delta}\right)/T + d \log\left(\frac{T}{\delta}\right)\sqrt{C/T}\right)$, matching the lower rate in [Arjevani et al., 2023] up to logarithm factors.*
*2) The convergence rate is of order $\tilde{\mathcal{O}}(1/T + \sqrt{C/T})$, and when the noise level $C$ is sufficiently low, the convergence rate could be $\tilde{\mathcal{O}}(1/T)$, which aligns with the result for non-adaptive SGD under the same conditions [Ghadimi and Lan, 2013, Bottou et al., 2018] up to logarithmic terms.*
*3) As in standard probability theory, the derived high-probability convergence can ensure expected convergence.*
*4) Assumption (A3) can be replaced by its sub-Gaussian form where $\mathbb{E}_{\boldsymbol{z}}\left[\exp\left(\frac{\|g(\boldsymbol{x},\boldsymbol{z}) - \nabla f(\boldsymbol{x})\|^2}{A(f(\boldsymbol{x}) - f^*) + B\|\nabla f(\boldsymbol{x})\|^2 + C}\right)\right] \leq \mathrm{e}$, and our results still hold true, as shown in Appendix.*

## 5 PROOF DETAIL

To start with, we let $\boldsymbol{g}_s = (g_{s,i})_i$ be as in Algorithm 1 and let $\nabla f(\boldsymbol{x}_s) = \bar{\boldsymbol{g}}_s = (\bar{g}_{s,i})_i$, $\boldsymbol{\xi}_s = (\xi_{s,i})_i = \boldsymbol{g}_s - \bar{\boldsymbol{g}}_s$ and $\Delta_s^{(x)} = f(\boldsymbol{x}_s) - f^*$.

During the proof, we will introduce several key lemmas to deduce the final results. All the missing proofs could be found in Appendix.

### 5.1 PRELIMINARY

Before proving the main result, we shall introduce several useful auxiliary sequences. The first sequence $\{\boldsymbol{y}_s\}_{s \geq 1}$ is defined as

$$\boldsymbol{y}_1 = \boldsymbol{x}_1, \boldsymbol{y}_s = \frac{\beta}{1 - \beta}(\boldsymbol{x}_s - \boldsymbol{x}_{s-1}) + \boldsymbol{x}_s, \quad \forall s \geq 2, \quad (7)$$

following from [Ghadimi et al., 2015, Yang et al., 2016] which was used to prove the convergence of SGD with momentum and later applied to handle with many variants of momentum-based algorithms. When $\boldsymbol{x}_s$ is generated by Algorithm 1, we reveal that $\boldsymbol{y}_s$ satisfies that for any $s \geq 1$,

$$\boldsymbol{y}_{s+1} = \boldsymbol{y}_s - \frac{\eta}{1 - \beta}\frac{\boldsymbol{g}_s}{\boldsymbol{b}_s}, \quad \boldsymbol{b}_s = \sqrt{\boldsymbol{v}_s} + \epsilon. \quad (8)$$

We let the function value gap $\Delta_s^{(y)} = f(\boldsymbol{y}_s) - f^*$. In addition, we introduce $\{\mathcal{G}_s\}_{s \geq 1}$ and the value $\mathcal{G}$,

$$\begin{aligned} \mathcal{G}_s &= \sqrt{X\Delta_s^{(x)} + 2C}, \\ \mathcal{G} &= \sqrt{X\Delta + 2C}, \quad X = 2A + 4LB + 4L, \end{aligned} \quad (9)$$

where $\Delta$ is as in Theorem 1.

### 5.2 ROUGH ESTIMATIONS

Motivated by [Faw et al., 2022], we provide some rough estimations for several key algorithm-dependent terms in this section. These estimations are not delicate, but they play vital roles in further deducing the final convergence rate.

**Lemma 5.1.** *For any $s \geq 1$ and $\beta \in [0, 1)$,*

$$\|\boldsymbol{m}_s\| \leq \frac{\eta\sqrt{d}}{1 - \beta}, \quad \|\bar{\boldsymbol{g}}_s\| \leq \|\bar{\boldsymbol{g}}_1\| + \frac{L\eta s\sqrt{d}}{1 - \beta}.$$

**Lemma 5.2.** *Suppose that $\beta \in [0, 1)$. Then for any $T \geq 1$,*

$$\sum_{t=1}^{T} \Delta_t^{(x)} \leq \Delta_1^{(x)} T$$
$$+ \left(\frac{\eta\|\bar{\boldsymbol{g}}_1\|\sqrt{d}}{1 - \beta} + \frac{L\eta^2 d}{2(1 - \beta)^2}\right)T^2 + \frac{L\eta^2 d T^3}{(1 - \beta)^2}.$$

### 5.3 START POINT AND DECOMPOSITION

We now proceed the proof for the main result. We fix the horizon $T$. Following [Ward et al., 2020], we start from the descent lemma of smoothness over $\boldsymbol{y}_s$ with both sides subtracting with $f^*$,

$$\Delta_{s+1}^{(y)} \leq \Delta_s^{(y)} + \langle\nabla f(\boldsymbol{y}_s), \boldsymbol{y}_{s+1} - \boldsymbol{y}_s\rangle + \frac{L}{2}\|\boldsymbol{y}_{s+1} - \boldsymbol{y}_s\|^2.$$

Combining with (8), and summing over $s \in [t]$,

$$\Delta_{t+1}^{(y)} \leq \Delta_1^{(x)} + \frac{\eta}{1-\beta} \underbrace{\left( -\sum_{s=1}^{t} \left\langle \nabla f(\boldsymbol{y}_s), \frac{\boldsymbol{g}_s}{\boldsymbol{b}_s} \right\rangle \right)}_{\mathbf{A}}$$
$$+ \frac{L\eta^2}{2(1-\beta)^2} \sum_{s=1}^{t} \left\| \frac{\boldsymbol{g}_s}{\boldsymbol{b}_s} \right\|^2, \quad (10)$$

where we apply $\boldsymbol{y}_1 = \boldsymbol{x}_1$. We subsequently further make a decomposition over $\mathbf{A}$ as

$$\mathbf{A} = \underbrace{-\sum_{s=1}^{t} \left\langle \bar{\boldsymbol{g}}_s, \frac{\boldsymbol{g}_s}{\boldsymbol{b}_s} \right\rangle}_{\mathbf{A.1}} + \underbrace{\sum_{s=1}^{t} \left\langle \bar{\boldsymbol{g}}_s - \nabla f(\boldsymbol{y}_s), \frac{\boldsymbol{g}_s}{\boldsymbol{b}_s} \right\rangle}_{\mathbf{A.2}}. \quad (11)$$

### 5.4 ESTIMATING A

The first main challenge comes from the entanglement of $\boldsymbol{g}_s$ and $\boldsymbol{b}_s$ emerging in $\mathbf{A}$, which is a key problem distinct from the analysis for SGD.

**Estimating A.1** We adopt the so-called proxy step-size technique which is a commonly used technique for breaking the correlation of $\boldsymbol{b}_s$ and $\boldsymbol{g}_s$ in the analysis of adaptive methods. This technique relies on introducing appropriate proxy step-sizes. It has been first introduced in [Ward et al., 2020] for AdaGrad-Norm with bounded stochastic gradients and variants of proxy step-sizes have been developed in the related literature, e.g., [Défossez et al., 2022, Faw et al., 2022, Attia and Koren, 2023, Liu et al., 2023b]. However, none of these proxy step-sizes could be potentially applied for AdaGrad with potential unbounded gradients under the mild noise model in Assumption (A3).

We thus provide a construction of proxy step-sizes that is general enough to handle with Assumption (A3). The proxy step-sizes rely on $\mathcal{G}_s$ given in (9), specifically defined in terms of

$$\boldsymbol{a}_s = \sqrt{\boldsymbol{v}_{s-1} + (\mathcal{G}_s \mathbf{1}_d)^2} + \boldsymbol{\epsilon}, \quad \forall s \in [T]. \quad (12)$$

Based on the proxy step-size $\eta / \boldsymbol{a}_s$, we further have

$$\mathbf{A.1} = -\sum_{s=1}^{t} \left\| \frac{\bar{\boldsymbol{g}}_s}{\sqrt{\boldsymbol{a}_s}} \right\|^2$$
$$\underbrace{-\sum_{s=1}^{t} \left\langle \bar{\boldsymbol{g}}_s, \frac{\boldsymbol{\xi}_s}{\boldsymbol{a}_s} \right\rangle}_{\mathbf{A.1.1}} + \underbrace{\sum_{s=1}^{t} \left\langle \bar{\boldsymbol{g}}_s, \left( \frac{1}{\boldsymbol{a}_s} - \frac{1}{\boldsymbol{b}_s} \right) \boldsymbol{g}_s \right\rangle}_{\mathbf{A.1.2}}. \quad (13)$$

The estimation for $\mathbf{A.1.1}$ relies on a probabilistic analysis over a summation of martingale difference sequence.

**Lemma 5.3.** *Given $T \geq 1$ and $\delta \in (0, 1)$, if Assumptions (A2) and (A3) hold, then with probability at least $1 - \delta$,*

$$\mathbf{A.1.1} \leq \frac{1}{4} \sum_{s=1}^{t} \frac{\mathcal{G}_s}{\mathcal{G}} \left\| \frac{\bar{\boldsymbol{g}}_s}{\sqrt{\boldsymbol{a}_s}} \right\|^2 + 3\mathcal{G} \log\left(\frac{T}{\delta}\right), \forall t \in [T], \quad (14)$$

*where $\mathcal{G}_s, \mathcal{G}$ are as in* (9).

The $\mathbf{A.1.2}$ serves as an error term for introducing $\boldsymbol{a}_s$. However, due to the delicate construction of $\boldsymbol{a}_s$, we could estimate the gap as follows,

**Lemma 5.4.** *Under Assumption (A3), let $\boldsymbol{b}_s = (b_{s,i})_i$, $\boldsymbol{a}_s = (a_{s,i})_i$ be defined in* (8) *and* (12). *Then*

$$\left| \frac{1}{a_{s,i}} - \frac{1}{b_{s,i}} \right| \leq \frac{\mathcal{G}_s}{a_{s,i} b_{s,i}}, \quad \forall s \in [T], \forall i \in [d].$$

Based on this lemma, it's then shown in the following lemma that $\mathbf{A.1.2}$ could be controlled.

**Lemma 5.5.** *Under Assumption (A3), for any $t \geq 1$, if $\beta \in [0, 1)$, it holds that*

$$\mathbf{A.1.2} \leq \frac{1}{4} \sum_{s=1}^{t} \left\| \frac{\bar{\boldsymbol{g}}_s}{\sqrt{\boldsymbol{a}_s}} \right\|^2 + \sum_{s=1}^{t} \mathcal{G}_s \left\| \frac{\boldsymbol{g}_s}{\boldsymbol{b}_s} \right\|^2. \quad (15)$$

Finally, we rely on the smoothness to estimate $\mathbf{A.2}$.

**Lemma 5.6.** *For any $t \geq 1$, if $\beta \in [0, 1)$, it holds that*

$$\mathbf{A.2} \leq \frac{L}{2\eta} \sum_{s=1}^{t} \|\boldsymbol{m}_{s-1}\|^2 + \frac{L\eta}{2(1-\beta)^2} \sum_{s=1}^{t} \left\| \frac{\boldsymbol{g}_s}{\boldsymbol{b}_s} \right\|^2. \quad (16)$$

### 5.5 BOUNDING THE FUNCTION VALUE GAP

Based on the above estimations, we could use an induction argument to deduce an upper bound for function value gaps. The induction technique is motivated by [Attia and Koren, 2023] where AdaGrad-Norm with affine variance noise was studied. As we study a more relaxed assumption on AdaGrad, it's required to provide some new estimations.

**Proposition 5.1.** *Under the same conditions of Theorem 1, the following two inequalities hold with probability at least $1 - \delta$,*

$$\Delta_t^{(x)} \leq \Delta, \quad \mathcal{G}_t \leq \mathcal{G}, \quad \forall t \in [T+1], \quad (17)$$

*and*

$$\Delta_{t+1}^{(x)} \leq \Delta - \frac{\eta}{1-\beta} \sum_{s=1}^{t} \left\| \frac{\bar{\boldsymbol{g}}_s}{\sqrt{\boldsymbol{a}_s}} \right\|^2, \quad \forall t \in [T], \quad (18)$$

*where $\Delta$ is as in Theorem 1 and $\mathcal{G}_t, \mathcal{G}$ are as in* (9).

In what follows, we prove Proposition 5.1. We assume that (14) always happens and then deduce (17) and (18). Recall that (14) holds with probability at least $1 - \delta$. We therefore obtain that both (17) and (18) would hold with probability at least $1 - \delta$. We first plug (13), (14) and (15) into (11), and then combine with (16) and (10) to get that

$$
\begin{aligned}
\Delta_{t+1}^{(y)} \leq{} & \Delta_1^{(x)} + \frac{\eta}{1-\beta} \sum_{s=1}^t \left( \frac{\mathcal{G}_s}{4\mathcal{G}} - \frac{3}{4} \right) \left\| \frac{\bar{\boldsymbol{g}}_s}{\sqrt{\boldsymbol{a}_s}} \right\|^2 \\
& + \frac{3\mathcal{G}\eta}{1-\beta} \log\left( \frac{T}{\delta} \right) + \frac{\eta}{1-\beta} \sum_{s=1}^t \mathcal{G}_s \left\| \frac{\boldsymbol{g}_s}{\boldsymbol{b}_s} \right\|^2 \\
& + \frac{L}{2(1-\beta)} \sum_{s=1}^t \|\boldsymbol{m}_{s-1}\|^2 + \tilde{L} \sum_{s=1}^t \left\| \frac{\boldsymbol{g}_s}{\boldsymbol{b}_s} \right\|^2, \quad (19)
\end{aligned}
$$

where we let $\tilde{L} = \frac{L\eta^2}{2(1-\beta)^3} + \frac{L\eta^2}{2(1-\beta)^2}$. Then, we present the specific definition of $\Delta$ as

$$
\begin{aligned}
\Delta :={} & 4\Delta_1^{(x)} + \frac{12\sqrt{2C}\eta}{1-\beta} \log\left( \frac{T}{\delta} \right) \quad (20) \\
& + 4\left( \frac{\sqrt{2C}\eta}{1-\beta} + \frac{\eta^2 L}{(1-\beta)^3} + \tilde{L} \right) d \log \mathcal{F}_T \\
& + \frac{72X\eta^2}{(1-\beta)^2} \log^2\left( \frac{T}{\delta} \right) + \frac{8X\eta^2}{(1-\beta)^2} d^2 \log^2 \mathcal{F}_T.
\end{aligned}
$$

Here, $\mathcal{F}_T$ is a polynomial with respect to $T$ with the detailed expression in (41) from Appendix. Then, it's easy to verify that $\Delta_1^{(x)} \leq \Delta$. Suppose that for some $t \in [T]$,

$$
\Delta_s^{(x)} \leq \Delta, \forall s \in [t], \quad \text{thus,} \quad \mathcal{G}_s \leq \mathcal{G}, \forall s \in [t]. \quad (21)
$$

In order to apply (19) to control $\Delta_{t+1}^{(x)}$, we introduce the following lemma to lower bound the LHS of (19).

**Lemma 5.7.** *Let $\boldsymbol{y}_s$ be defined in* (7) *and $\beta \in [0,1)$. Then for any $s \geq 1$,*

$$
\Delta_s^{(y)} \geq \frac{\Delta_s^{(x)}}{2} - \frac{L\|\boldsymbol{m}_{s-1}\|^2}{2(1-\beta)^2}.
$$

Based on Lemma 5.7, the LHS of (19) could be lower bounded in terms of $\Delta_{t+1}^{(x)}$. We use (21) to upper bound the RHS of (19), which leads to

$$
\begin{aligned}
\frac{\Delta_{t+1}^{(x)}}{2} \leq{} & \Delta_1^{(x)} - \frac{\eta}{2(1-\beta)} \sum_{s=1}^t \left\| \frac{\bar{\boldsymbol{g}}_s}{\sqrt{\boldsymbol{a}_s}} \right\|^2 + \frac{L\|\boldsymbol{m}_t\|^2}{2(1-\beta)^2} \\
& + \frac{3(\sqrt{X\Delta} + \sqrt{2C})\eta}{1-\beta} \log\left( \frac{T}{\delta} \right) + \tilde{L} \sum_{s=1}^t \left\| \frac{\boldsymbol{g}_s}{\boldsymbol{b}_s} \right\|^2 \\
& + \frac{(\sqrt{X\Delta} + \sqrt{2C})\eta}{1-\beta} \sum_{s=1}^t \left\| \frac{\boldsymbol{g}_s}{\boldsymbol{b}_s} \right\|^2 + \sum_{s=1}^t \frac{L\|\boldsymbol{m}_{s-1}\|^2}{2(1-\beta)},
\end{aligned}
$$

where we use $\mathcal{G} \leq \sqrt{X\Delta} + \sqrt{2C}$. Further, using Young's inequality twice for the terms related to $\sqrt{X\Delta}$, and $\beta < 1$,

$$
\begin{aligned}
\frac{\Delta_{t+1}^{(x)}}{2} \leq{} & \frac{\Delta}{4} + \Delta_1^{(x)} - \frac{\eta}{2(1-\beta)} \sum_{s=1}^t \left\| \frac{\bar{\boldsymbol{g}}_s}{\sqrt{\boldsymbol{a}_s}} \right\|^2 \\
& + \frac{L\|\boldsymbol{m}_t\|^2}{2(1-\beta)^2} + \frac{3\sqrt{2C}\eta}{1-\beta} \log\left( \frac{T}{\delta} \right) \\
& + \left( \frac{\sqrt{2C}\eta}{1-\beta} + \tilde{L} \right) \sum_{s=1}^t \left\| \frac{\boldsymbol{g}_s}{\boldsymbol{b}_s} \right\|^2 + \sum_{s=1}^t \frac{L\|\boldsymbol{m}_{s-1}\|^2}{2(1-\beta)} \\
& + \frac{18X\eta^2}{(1-\beta)^2} \log^2\left( \frac{T}{\delta} \right) + \frac{2X\eta^2}{(1-\beta)^2} \left( \sum_{s=1}^t \left\| \frac{\boldsymbol{g}_s}{\boldsymbol{b}_s} \right\|^2 \right)^2.
\end{aligned}
$$
(22)

Finally, we shall use the following lemma to further estimate $\|\boldsymbol{m}_t\|$ and the other two summations related to $\boldsymbol{g}_s, \boldsymbol{b}_s, \boldsymbol{m}_s$.

**Lemma 5.8.** *Given $T \geq 1$ and $\beta \in [0,1)$, then for any $t \in [T]$,*

$$
\sum_{s=1}^t \left\| \frac{\boldsymbol{g}_s}{\boldsymbol{b}_s} \right\|^2 \leq d \log \mathcal{F}_T, \quad \|\boldsymbol{m}_t\|^2 \leq \frac{\eta^2 d}{1-\beta} \log \mathcal{F}_T,
$$

$$
\sum_{s=1}^t \|\boldsymbol{m}_s\|^2 \leq \frac{\eta^2 d}{(1-\beta)^2} \log \mathcal{F}_T,
$$

*where $\mathcal{F}_T$ is a polynomial with respect to $T$ with the detailed expression in* (41) *from Appendix.*

Compared with Lemma 5.1, Lemma 5.8 improves the dependency to $1 - \beta$ for estimating $\|\boldsymbol{m}_t\|^2$, which leads to the $\mathcal{O}((1-\beta)^{-1})$ order for the final convergence as in Remark 4.1. Thus, applying Lemma 5.8 over (22), and then combining with $\Delta$ in (20),

$$
\frac{\Delta_{t+1}^{(x)}}{2} \leq \frac{\Delta}{2} - \frac{\eta}{2(1-\beta)} \sum_{s=1}^t \left\| \frac{\bar{\boldsymbol{g}}_s}{\sqrt{\boldsymbol{a}_s}} \right\|^2 \leq \frac{\Delta}{2}.
$$

The induction is complete and the desired result in (17) is proved. Finally, as an intermediate result, we verify (18).

## 5.6 PROOF OF THE MAIN RESULT

Based on Proposition 5.1, we are able to prove Theorem 1.

*Proof of Theorem 1.* In what follows, we will obtain the final convergence result based on (17) and (18). Since (17) and (18) hold with probability at least $1 - \delta$, the final convergence result then holds with probability at least $1 - \delta$. Let us first set $t = T$ in (18), and we get

$$
\frac{\eta}{1-\beta} \sum_{s=1}^T \frac{\|\bar{\boldsymbol{g}}_s\|^2}{\|\boldsymbol{a}_s\|_\infty} \leq \frac{\eta}{1-\beta} \sum_{s=1}^T \left\| \frac{\bar{\boldsymbol{g}}_s}{\sqrt{\boldsymbol{a}_s}} \right\|^2 \leq \Delta. \quad (23)
$$

Using (12), the basic inequality and Assumption (A3), we have that for any $s \in [T]$, with $B_1 = B + 1$,

$$\|\boldsymbol{a}_s\|_\infty - \epsilon \le \max_{i \in [d]} \sqrt{v_{s-1,i} + \mathcal{G}_s^2} = \max_{i \in [d]} \sqrt{\sum_{j=1}^{s-1} g_{j,i}^2 + \mathcal{G}_s^2}$$

$$\le \sqrt{\sum_{j=1}^{s-1} \|\boldsymbol{g}_j\|^2 + \mathcal{G}_s^2} \le \sqrt{2 \sum_{j=1}^{s-1} (\|\bar{\boldsymbol{g}}_j\|^2 + \|\boldsymbol{\xi}_j\|^2) + \mathcal{G}_s^2}$$

$$\le \sqrt{2 \sum_{j=1}^{s-1} (A\Delta_j^{(x)} + B_1 \|\bar{\boldsymbol{g}}_j\|^2 + C) + \mathcal{G}_s^2}. \quad (24)$$

Further applying (17) where $\Delta_s^{(x)} \le \Delta, \mathcal{G}_s \le \mathcal{G}, \forall s \in [T]$,

$$\|\boldsymbol{a}_s\|_\infty - \epsilon \le \sqrt{2B_1 \sum_{s=1}^{T} \|\bar{\boldsymbol{g}}_s\|^2 + 2(A\Delta + C)T + \mathcal{G}^2}.$$

Combining with (23), using $\Delta_1 = \Delta(1-\beta)/\eta$, then applying Young's inequality,

$$\sum_{s=1}^{T} \|\bar{\boldsymbol{g}}_s\|^2 - \Delta_1 \epsilon$$

$$\le \Delta_1 \left( \sqrt{2B_1 \sum_{s=1}^{T} \|\bar{\boldsymbol{g}}_s\|^2} + \sqrt{2(A\Delta + C)T} + \mathcal{G} \right)$$

$$\le \sum_{s=1}^{T} \frac{\|\bar{\boldsymbol{g}}_s\|^2}{2} + \Delta_1^2 B_1 + \Delta_1 \left( \sqrt{2(A\Delta + C)T} + \mathcal{G} \right).$$

We then re-arrange the order and divide $T$ on both sides, leading to a desired convergence result

$$\frac{1}{T} \sum_{s=1}^{T} \|\bar{\boldsymbol{g}}_s\|^2 \le 2\Delta_1 \left[ \frac{\Delta_1 B_1 + \mathcal{G} + \epsilon}{T} + \sqrt{\frac{2(A\Delta + C)}{T}} \right].$$

The proof is complete. □

# 6 CONVERGENCE UNDER GENERALIZED SMOOTHNESS

In this section, we present the convergence of Algorithm 1 in the generalized smooth case.

## 6.1 GENERALIZED SMOOTHNESS

For a differentiable objective function $f : \mathbb{R}^d \to \mathbb{R}$, we consider the following $(L_0, L_1)$-smoothness condition: there exist constants $L_0, L_1 > 0$, satisfying that for any $\boldsymbol{x}, \boldsymbol{y} \in \mathbb{R}^d$ with $\|\boldsymbol{x} - \boldsymbol{y}\| \le 1/L_1$,

$$\|\nabla f(\boldsymbol{y}) - \nabla f(\boldsymbol{x})\| \le (L_0 + L_1 \|\nabla f(\boldsymbol{x})\|) \|\boldsymbol{x} - \boldsymbol{y}\|. \quad (25)$$

The generalized smooth condition was originally put forward by [Zhang et al., 2020b] for any twice-order differentiable function $f$ satisfying that

$$\|\nabla^2 f(\boldsymbol{x})\| \le L_0 + L_1 \|\nabla f(\boldsymbol{x})\|. \quad (26)$$

They revealed the superior of SGD with gradient-clipping in convergence over the vanilla SGD when considering (26). Moreover, empirical evidence has demonstrated that numerous objective functions satisfy (26) while deviating from the global smoothness, particularly in large language models, see e.g., [Zhang et al., 2020b, Figure 1] and [Crawshaw et al., 2022]. To better explain the convergence of gradient-clipping algorithms, Zhang et al. [2020a] provided an alternative form in (25), only requiring $f$ to be first-order differentiable.

The condition in (25) is selected in our paper for three main reasons. First, it's easy to verify that (25) is strictly weaker than $L$-smoothness. A concrete example is that the simple function $f(x) = x^3, x \in \mathbb{R}$ does not satisfy any global smoothness but (25). Second, (25) aligns with the practical limitation to first-order stochastic gradients in our setting, making it more reasonable to assume that $f$ is only first-order differentiable. Finally, both (25) and (26) are shown to be equivalent up to constant factors for twice-order differentiable functions, see [Zhang et al., 2020a, Lemma A.2] and [Faw et al., 2023, Proposition 1]. Thus, (25) includes more functions than (26). We refer interested readers to see [Zhang et al., 2020b,a, Faw et al., 2023] for more discussions and concrete examples of the generalized smoothness.

## 6.2 CONVERGENCE RESULT

In the following, we establish the convergence bound for AdaGrad with momentum under the generalized smooth condition.

**Theorem 2.** *Let $T \ge 1$ and $\delta \in (0,1)$. Suppose that $\{\boldsymbol{x}_s\}_{s \in [T]}$ is a sequence generated by Algorithm 1, $f$ is $(L_0, L_1)$-smooth satisfying (25), Assumptions (A1), (A2), (A3) hold, and the parameters satisfy that $\beta \in [0, 1)$,*

$$\epsilon > 0, \quad \eta \le \min \left\{ C_0, \frac{C_0}{\mathcal{H}}, \frac{C_0}{\mathcal{L}}, \frac{(1-\beta)^2}{L_1 \sqrt{d}} \right\}, \quad (27)$$

*where $C_0 > 0$ is a constant, $\mathcal{H}, \mathcal{L}$ are defined as*

$$\mathcal{H} = \sqrt{2A\Lambda_x + 2(B+1) \left( 4L_1 \Lambda_x + \sqrt{4L_0 \Lambda_x} \right)^2 + 2C},$$

$$\mathcal{L} = 2L_0 + 2L_1 \left( 4L_1 \Lambda_x + \sqrt{4L_0 \Lambda_x} \right),$$

$$(28)$$

*and $\Lambda_y, \tilde{\Lambda}_y, \Lambda_x$ are given with the following order,[4]*

$$\Lambda_y \sim \mathcal{O}\left(\Delta_1^{(x)} + \frac{C_0^2 d + C_0 d}{(1-\beta)^3} \log\left(\frac{T}{\delta} + \frac{T}{\epsilon^2}\right)\right),$$

$$\tilde{\Lambda}_y = \frac{2\Lambda_y(1-\beta)}{\eta}, \quad \Lambda_x \sim \mathcal{O}\left(\Lambda_y^2\right).$$

*Then, with $B_1 = B + 1$, it holds that with probability at least $1 - \delta$,*

$$\frac{1}{T}\sum_{s=1}^{T}\|\nabla f(\boldsymbol{x}_s)\|^2$$

$$\leq \mathcal{O}\left(\tilde{\Lambda}_y\left(\frac{B_1\tilde{\Lambda}_y + \sqrt{B_1 L \Lambda_x} + \epsilon}{T} + \sqrt{\frac{A\Lambda_x + C}{T}}\right)\right).$$

It's easy to verify that $\Lambda_y \sim \mathcal{O}(\log(T/\delta))$ and thereby $\Lambda_x, \mathcal{H}, \mathcal{L} \sim \mathcal{O}(\log^2(T/\delta))$. Then, from (27), when $T \gg d$, a typical setting is $\eta \sim \mathcal{O}(\log^2(T/\delta))$. Moreover, the convergence rate is still adaptive to the noise parameters $A, C$ and requires problem-parameters to tune step-sizes potentially due to the relaxation of smoothness. The subsequent result for AdaGrad under (25) could be directly deduced from Theorem 2 and will be presented in Appendix.

# 7 CONCLUSION

In this paper, we provide high probability convergence bounds for AdaGrad and its momentum variant under the non-convex smooth optimization. In particular, we consider a mild noise model incorporating affine variance noise and the expected smoothness. We rely on a new proxy step-size and some delicate estimations to derive the bound. Our findings reveal that without problem-parameters dependent step-sizes, AdaGrad can find a stationary point with a rate of $\tilde{\mathcal{O}}(1/\sqrt{T})$, particularly accelerating to $\tilde{\mathcal{O}}(1/T)$ when specific noise parameters are sufficiently small. Furthermore, we extend our framework to the generalized smooth case that allows for unbounded smooth parameters, showing the same convergence rate, albeit that problem-parameters dependent step-sizes are required in the latter.

**Limitation**  Although AdaGrad plays an important role in the adaptive method field, several other adaptive methods including RMSProp, Adam and AdamW, may be preferred in some real applications. Therefore, it is also pertinent to study these algorithms under relaxed assumptions. In addition, it is still unknown whether similar convergence result could be also achieved under an expected version of Assumption (A3). Finally, as we study a new assumption over AdaGrad, it would be more beneficial to provide more experimental results to support the theoretical results.

---

[4]The detailed expressions of $\Lambda_y, \Lambda_x$ could be found in (72) and (73) respectively from Appendix.

# ACKNOWLEDGMENTS

The authors would like to thank the reviewers and area chairs for their constructive comments. This work was supported in part by the National Key Research and Development Program of China under grant number 2021YFA1003500, and NSFC under grant numbers 11971427. We also thank Chenhao Yu very much on pointing out one error in the generalized-smooth analysis, and Mouxiang Chen for his great help with the experimental results. The corresponding author is Junhong Lin.

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

# Revisiting Convergence of AdaGrad with Relaxed Assumptions
## (Supplementary Material)

**Yusu Hong**[1]                    **Junhong Lin**[1]

[1]Zhejiang University

Table 1: Comparison of existing results with ours for AdaGrad/AdaGrad-Norm on non-convex smooth case

|  | Alg. type | Smooth | Noise | Unbounded Gradients | Conv. type |
|---|---|---|---|---|---|
| [Li and Orabona, 2019] | Both[a] | $L$ | Sub-Gaussian | - | w.h.p. |
| [Ward et al., 2020] | Scalar | $L$ | Bounded | - | $\mathbb{E}$ |
| [Kavis et al., 2022] | Scalar | $L$ | Sub-Gaussian | ✓ | w.h.p. |
| [Faw et al., 2022] | Scalar | $L$ | Affine | ✓ | $\mathbb{E}$ |
| [Wang et al., 2023] | Both | $L/(L_0, L_1)$ | Affine | ✓ | $\mathbb{E}$ |
| [Liu et al., 2023b] | Both | $L$ | Coordinate-wise Sub-Gaussian | ✓ | w.h.p. |
| [Attia and Koren, 2023] | Scalar | $L$ | Affine | ✓ | w.h.p. |
| [Faw et al., 2023] | Scalar | $(L_0, L_1)$ | Affine | ✓ | $\mathbb{E}$ |
| This paper, Thm. 1 | Coordinate | $L$ | Relaxed Affine | ✓ | w.h.p. |
| This paper, Thm. 2 | Coordinate | $(L_0, L_1)$ | Relaxed Affine | ✓ | w.h.p. |

In the "Alg. type" column, "Scalar" refers to AdaGrad-Norm, "Coordinate" refers to AdaGrad, and "Both" refers to both algorithms. "Relaxed Affine" corresponds to Assumption (A3) in this paper. In the "Conv. type" column, "w.h.p." stands for the high probability convergence bound, and "$\mathbb{E}$" represents the expected convergence bound.

[a] Li and Orabona [2019] studied a variant of AdaGrad/AdaGrad-Norm using a delayed step-size which is independent from the current stochastic gradient.

## A    COMPLEMENTARY LEMMAS

Following [Li and Orabona, 2020, Attia and Koren, 2023], we will first present several important technical lemmas. The first lemma is a standard result in the smooth-based optimization which will be used in our analysis motivated also by [Attia and Koren, 2023, Hong and Lin, 2023].

**Lemma A.1.** *Suppose that $f$ is L-smooth and Assumption (A1) holds. Then for any $\boldsymbol{x} \in \mathbb{R}^d$,*

$$\|\nabla f(\boldsymbol{x})\|^2 \leq 2L(f(\boldsymbol{x}) - f^*).$$

We introduce a concentration inequality for the martingale difference sequence, see [Li and Orabona, 2020] for a proof.

**Lemma A.2.** *Suppose that $\{Z_s\}_{s \in [T]}$ is a martingale difference sequence with respect to $\zeta_1, \cdots, \zeta_T$. Assume that for each $s \in [T]$, $\sigma_s$ is a random variable dependent on $\zeta_1, \cdots, \zeta_{s-1}$ and satisfies that*

$$\mathbb{E}\left[\exp\left(\frac{Z_s^2}{\sigma_s^2}\right) \mid \zeta_1, \cdots, \zeta_{s-1}\right] \leq \mathrm{e}.$$

*Accepted for the 40[th] Conference on Uncertainty in Artificial Intelligence* (UAI 2024).

*Then, for any $\lambda > 0$, and for any $\delta \in (0, 1)$, it holds that*

$$\mathbb{P}\left(\sum_{s=1}^{T} Z_s > \frac{1}{\lambda}\log\left(\frac{1}{\delta}\right) + \frac{3}{4}\lambda\sum_{s=1}^{T}\sigma_s^2\right) \le \delta.$$

The following lemma is a commonly used result in the analysis of adaptive methods. See [Levy et al., 2018] for a proof.

**Lemma A.3.** *Let $\{\alpha_s\}_{s\ge 1}$ be a non-negative sequence. For any $\varepsilon > 0$ and positive integer $t$,*

$$\sum_{s=1}^{t}\frac{\alpha_s}{\varepsilon + \sum_{k=1}^{s}\alpha_k} \le \log\left(1 + \frac{1}{\varepsilon}\sum_{s=1}^{t}\alpha_s\right).$$

# B   OMITTED PROOFS UNDER SMOOTH CASE

In this section, we provide the missing detailed proofs for some results and lemmas used in the proof of Theorem 1.

*Proof of Remark 4.2.* Here, we prove the fourth point in Remark 4.2. Let us fix horizon $T$ and denote $\gamma_s = \frac{\|\boldsymbol{g}_s - \bar{\boldsymbol{g}}_s\|^2}{A\Delta_s^{(x)} + B\|\bar{\boldsymbol{g}}_s\|^2 + C}, \forall s \in [T]$. Then from the new assumption, we have

$$\mathbb{E}_{\boldsymbol{z}_s}[\exp(\gamma_s)] \le \mathrm{e}, \quad \text{thus,} \quad \mathbb{E}[\exp(\gamma_s)] \le \mathrm{e}.$$

By Markov's inequality, for any $E \in \mathbb{R}$,

$$\mathbb{P}\left(\max_{s\in[T]}\gamma_s \ge E\right) = \mathbb{P}\left(\exp\left(\max_{s\in[T]}\gamma_s\right) \ge \mathrm{e}^E\right) \le \mathrm{e}^{-E}\mathbb{E}\left[\exp\left(\max_{s\in[T]}\gamma_s\right)\right] \le \mathrm{e}^{-E}\mathbb{E}\left[\sum_{s=1}^{T}\exp(\gamma_s)\right] \le \mathrm{e}^{-E}T\mathrm{e},$$

which leads to that with probability at least $1 - \delta$,

$$\|\boldsymbol{\xi}_s\|^2 = \|\boldsymbol{g}_s - \bar{\boldsymbol{g}}_s\|^2 \le \log\left(\frac{\mathrm{e}T}{\delta}\right)\left(A\Delta_s^{(x)} + B\|\bar{\boldsymbol{g}}_s\|^2 + C\right), \quad \forall s \in [T]. \tag{29}$$

Compared (29) with Assumption (A3), an additional $\log T$ factor emerges. Hence, $\mathcal{G}_s$ and $\mathcal{G}$ defined in (9) should be revised as

$$\mathcal{G}_s = \sqrt{\left(X\Delta_s^{(x)} + 2C\right)\log\left(\frac{\mathrm{e}T}{\delta}\right)}, \quad \mathcal{G} = \sqrt{(X\Delta + 2C)\log\left(\frac{\mathrm{e}T}{\delta}\right)}. \tag{30}$$

Consequently, using the similar analysis in Lemma 5.3, Lemma 5.5 and Lemma 5.6, we could reach (19) with a new $\mathcal{G}_s$ and $\mathcal{G}$ in (30). Then, using a similar induction argument, we could deduce the final convergence rate. The additional logarithm factor will make no essential difference to the order of $\Delta, \Delta_1$ and the convergence rate in Theorem 1 up to logarithm factors. $\qquad\square$

*Proof of Lemma 5.1.* Let us denote $\boldsymbol{\eta}_s = \eta/(\sqrt{\boldsymbol{v}_s} + \boldsymbol{\epsilon})$. Recalling $\boldsymbol{m}_0 = 0$ and $\boldsymbol{m}_s$ in Algorithm 1, we have

$$\boldsymbol{m}_s = \beta\boldsymbol{m}_{s-1} - \boldsymbol{\eta}_s \odot \boldsymbol{g}_s = \beta^2\boldsymbol{m}_{s-2} - \beta\boldsymbol{\eta}_{s-1}\odot\boldsymbol{g}_{s-1} - \boldsymbol{\eta}_s\odot\boldsymbol{g}_s = \cdots = -\sum_{j=1}^{s}\beta^{s-j}\boldsymbol{\eta}_j\odot\boldsymbol{g}_j. \tag{31}$$

Note that $|g_{s,i}| \le \sqrt{v_{s,i}}, \forall i \in [d]$. We therefore verify that $\|\boldsymbol{g}_s/\sqrt{\boldsymbol{v}_s}\| \le \sqrt{d}\max_{i\in[d]}|g_{s,i}/v_{s,i}| \le \sqrt{d}$. Moreover,

$$\|\boldsymbol{m}_s\| \le \sum_{j=1}^{s}\beta^{s-j}\|\boldsymbol{\eta}_j\odot\boldsymbol{g}_j\| \le \eta\sqrt{d}\sum_{j=1}^{s}\beta^{s-j}\left\|\frac{\boldsymbol{g}_j}{\sqrt{\boldsymbol{v}_j}}\right\|_{\infty} \le \frac{\eta\sqrt{d}}{1-\beta}. \tag{32}$$

Using the smoothness of $f$,

$$\|\bar{\boldsymbol{g}}_s\| \le \|\bar{\boldsymbol{g}}_{s-1}\| + \|\bar{\boldsymbol{g}}_s - \bar{\boldsymbol{g}}_{s-1}\| \le \|\bar{\boldsymbol{g}}_{s-1}\| + L\|\boldsymbol{x}_s - \boldsymbol{x}_{s-1}\| = \|\bar{\boldsymbol{g}}_{s-1}\| + L\|\boldsymbol{m}_{s-1}\|.$$

Further using (32),

$$\|\bar{g}_s\| \le \|\bar{g}_{s-1}\| + \frac{L\eta\sqrt{d}}{1-\beta} \le \|\bar{g}_1\| + \frac{L\eta s\sqrt{d}}{1-\beta}.$$

$\square$

*Proof of Lemma 5.2.* Recalling the iteration in Algorithm 1 and then using the descent lemma,

$$f(\boldsymbol{x}_{s+1}) \le f(\boldsymbol{x}_s) + \langle \bar{g}_s, \boldsymbol{x}_{s+1} - \boldsymbol{x}_s \rangle + \frac{L}{2}\|\boldsymbol{x}_{s+1} - \boldsymbol{x}_s\|^2 = f(\boldsymbol{x}_s) + \langle \bar{g}_s, \boldsymbol{m}_s \rangle + \frac{L}{2}\|\boldsymbol{m}_s\|^2.$$

Using Cauchy-Schwarz inequality and Lemma 5.1,

$$\langle \bar{g}_s, \boldsymbol{m}_s \rangle \le \|\bar{g}_s\| \cdot \|\boldsymbol{m}_s\| \le \frac{\eta\sqrt{d}}{1-\beta}\left(\|\bar{g}_1\| + \frac{L\eta\sqrt{d}s}{1-\beta}\right), \quad \frac{L}{2}\|\boldsymbol{m}_s\|^2 \le \frac{L\eta^2 d}{2(1-\beta)^2}.$$

Combining with the above, we obtain that

$$f(\boldsymbol{x}_{s+1}) \le f(\boldsymbol{x}_s) + \frac{\eta\sqrt{d}}{1-\beta}\left(\|\bar{g}_1\| + \frac{L\eta\sqrt{d}s}{1-\beta}\right) + \frac{L\eta^2 d}{2(1-\beta)^2}.$$

With both sides subtracting $f^*$ and summing up over $s \in [t]$, we obtain that

$$\Delta_{t+1}^{(x)} \le \Delta_1^{(x)} + \frac{\eta\sqrt{d}}{1-\beta}\sum_{s=1}^{t}\left(\|\bar{g}_1\| + \frac{L\eta\sqrt{d}s}{1-\beta}\right) + \frac{L\eta^2 dt}{2(1-\beta)^2}. \tag{33}$$

We define $\sum_a^b = 0$ when $a < b$. Then, we sum up both sides of (33) over $t \in [0, 1, \cdots, T-1]$ to obtain that

$$\begin{aligned}
\sum_{t=1}^{T}\Delta_t^{(x)} &\le \sum_{t=0}^{T-1}\Delta_1^{(x)} + \frac{\eta\sqrt{d}}{1-\beta}\sum_{t=0}^{T-1}\sum_{s=1}^{t}\left(\|\bar{g}_1\| + \frac{L\eta\sqrt{d}s}{1-\beta}\right) + \sum_{t=0}^{T-1}\frac{L\eta^2 dt}{2(1-\beta)^2} \\
&\le \Delta_1^{(x)}T + \frac{\eta\sqrt{d}}{1-\beta}\sum_{t=1}^{T}\sum_{s=1}^{t}\left(\|\bar{g}_1\| + \frac{L\eta\sqrt{d}s}{1-\beta}\right) + \sum_{t=1}^{T}\frac{L\eta^2 dt}{2(1-\beta)^2} \\
&\le \Delta_1^{(x)}T + \left(\frac{\eta\|\bar{g}_1\|\sqrt{d}}{1-\beta} + \frac{L\eta^2 d}{2(1-\beta)^2}\right)T^2 + \frac{L\eta^2 dT^3}{(1-\beta)^2}.
\end{aligned}$$

$\square$

*Proof of Lemma 5.3.* Let $X_s = -\left\langle \bar{g}_s, \frac{\boldsymbol{\xi}_s}{\boldsymbol{a}_s} \right\rangle$ for any $s \in [T]$. Note that $\bar{g}_s$ and $\boldsymbol{a}_s$ are random variables dependent on $\boldsymbol{z}_1, \cdots, \boldsymbol{z}_{s-1}$ and $\boldsymbol{\xi}_s$ is dependent on $\boldsymbol{z}_1, \cdots, \boldsymbol{z}_{s-1}, \boldsymbol{z}_s$. It is easy to prove that $X_s$ is a martingale difference sequence since

$$\mathbb{E}\left[X_s \mid \boldsymbol{z}_1, \cdots, \boldsymbol{z}_{s-1}\right] = -\left\langle \bar{g}_s, \frac{\mathbb{E}_{\boldsymbol{z}_s}[\boldsymbol{\xi}_s]}{\boldsymbol{a}_s} \right\rangle = 0,$$

where the last equality follows from Assumption (A2). Let

$$\zeta_s = \left\|\frac{\bar{g}_s}{\boldsymbol{a}_s}\right\|\sqrt{A\Delta_s^{(x)} + B\|\bar{g}_s\|^2 + C}, \quad \forall s \in [T].$$

Similarly, $\zeta_s$ is a random variable dependent on $\boldsymbol{z}_1, \cdots, \boldsymbol{z}_{s-1}$. Using Cauchy-Schwarz inequality and Assumption (A3), we have

$$\mathbb{E}\left[\exp\left(\frac{X_s^2}{\zeta_s^2}\right) \mid \boldsymbol{z}_1, \cdots, \boldsymbol{z}_{s-1}\right] \le \mathbb{E}\left[\exp\left(\frac{\|\boldsymbol{\xi}_s\|^2}{A\Delta_s^{(x)} + B\|\bar{g}_s\|^2 + C}\right) \mid \boldsymbol{z}_1, \cdots, \boldsymbol{z}_{s-1}\right] \le e.$$

Therefore, given any fixed $t \in [T]$, applying Lemma A.2, we have that for any $\lambda > 0$, with probability at least $1 - \delta$,

$$\sum_{s=1}^{t} X_s^2 \leq \frac{3\lambda}{4} \sum_{s=1}^{t} \zeta_s^2 + \frac{1}{\lambda} \log\left(\frac{1}{\delta}\right) = \frac{3\lambda}{4} \sum_{s=1}^{t} \sum_{i=1}^{d} \frac{\bar{g}_{s,i}^2}{a_{s,i}^2} \left(A\Delta_s^{(x)} + B\|\bar{g}_s\|^2 + C\right) + \frac{1}{\lambda} \log\left(\frac{1}{\delta}\right)$$

$$\leq \frac{3\lambda}{4} \sum_{s=1}^{t} \sum_{i=1}^{d} \frac{\bar{g}_{s,i}^2 \mathcal{G}_s^2}{a_{s,i}^2} + \frac{1}{\lambda} \log\left(\frac{1}{\delta}\right) \leq \frac{3\lambda}{4} \sum_{s=1}^{t} \sum_{i=1}^{d} \frac{\bar{g}_{s,i}^2 \mathcal{G}_s}{a_{s,i}} + \frac{1}{\lambda} \log\left(\frac{1}{\delta}\right), \tag{34}$$

where the second inequality follows from Lemma A.1 and (9). The last inequality follows from using $1/a_{s,i} \leq 1/\mathcal{G}_s$, implied by (12). We then obtain that for any $t \in [T]$, (34) holds with probability at least $1 - \delta$. Therefore, for any fixed $\lambda > 0$, we can re-scale over $\delta$ and have that for all $t \in [T]$, with probability at least $1 - \delta$,

$$\sum_{s=1}^{t} X_s^2 \leq \frac{3\lambda}{4} \sum_{s=1}^{t} \left\| \frac{\bar{g}_s}{\sqrt{a_s}} \right\|^2 \mathcal{G}_s + \frac{1}{\lambda} \log\left(\frac{T}{\delta}\right). \tag{35}$$

Finally setting $\lambda = 1/(3\mathcal{G})$, we obtain the desired result. $\qquad\square$

*Proof of Lemma 5.4.* Using the basic inequality, Assumption (A3) and Lemma A.1, we have

$$\|g_s\|^2 \leq 2\|\bar{g}_s\|^2 + 2\|\xi_s\|^2 \leq 2A\Delta_s^{(x)} + 2(B+1)\|\bar{g}_s\|^2 + 2C \leq (2A + 4LB + 4L)\Delta_s^{(x)} + 2C.$$

Thus, $\|g_s\| \leq \mathcal{G}_s, \forall s \geq 1$. Let $a_s = \sqrt{\tilde{v}_s} + \epsilon$ where $\tilde{v}_s = (\tilde{v}_{s,i})_i$. Then, for any $i \in [d]$,

$$\left| \frac{1}{a_{s,i}} - \frac{1}{b_{s,i}} \right| = \frac{|\sqrt{\tilde{v}_{s,i}} - \sqrt{v_{s,i}}|}{a_{s,i} b_{s,i}} = \frac{|\tilde{v}_{s,i} - v_{s,i}|}{a_{s,i} b_{s,i} (\sqrt{\tilde{v}_{s,i}} + \sqrt{v_{s,i}})} = \frac{1}{a_{s,i} b_{s,i}} \frac{|\mathcal{G}_s^2 - g_{s,i}^2|}{\sqrt{v_{s,i}} + \sqrt{\tilde{v}_{s,i}}}$$

$$\leq \frac{1}{a_{s,i} b_{s,i}} \frac{\mathcal{G}_s^2}{\sqrt{v_{s,i}} + \sqrt{\tilde{v}_{s,i}}} \leq \frac{\mathcal{G}_s}{a_{s,i} b_{s,i}}. \tag{36}$$

$\qquad\square$

*Proof of Lemma 5.5.* Applying Lemma 5.4 and Young's inequality,

$$\mathbf{A.1.2} \leq \sum_{s=1}^{t} \sum_{i=1}^{d} \left| \frac{1}{a_{s,i}} - \frac{1}{b_{s,i}} \right| |\bar{g}_{s,i}||g_{s,i}| \leq \sum_{s=1}^{t} \sum_{i=1}^{d} \frac{\mathcal{G}_s}{a_{s,i} b_{s,i}} |\bar{g}_{s,i}||g_{s,i}| \leq \frac{1}{4} \sum_{s=1}^{t} \sum_{i=1}^{d} \frac{\bar{g}_{s,i}^2}{a_{s,i}} + \sum_{s=1}^{t} \sum_{i=1}^{d} \frac{\mathcal{G}_s^2}{a_{s,i}} \frac{g_{s,i}^2}{b_{s,i}^2}$$

$$\leq \frac{1}{4} \sum_{s=1}^{t} \sum_{i=1}^{d} \frac{\bar{g}_{s,i}^2}{a_{s,i}} + \sum_{s=1}^{t} \sum_{i=1}^{d} \mathcal{G}_s \frac{g_{s,i}^2}{b_{s,i}^2}, \tag{37}$$

where we use $1/a_{s,i} \leq 1/\mathcal{G}_s$ for the last inequality. The proof is complete. $\qquad\square$

*Proof of Lemma 5.7.* Using the descent lemma of smoothness and (7),

$$f(x_s) \leq f(y_s) + \langle \nabla f(y_s), x_s - y_s \rangle + \frac{L}{2} \|x_s - y_s\|^2$$

$$= f(y_s) - \frac{\beta}{1-\beta} \langle \nabla f(y_s), x_s - x_{s-1} \rangle + \frac{L\beta^2}{2(1-\beta)^2} \|x_s - x_{s-1}\|^2.$$

Using Young's inequality and subtracting $f^*$ on both sides,

$$\Delta_s^{(x)} \leq \Delta_s^{(y)} + \frac{1}{2L} \|\nabla f(y_s)\|^2 + \frac{(L+L)\beta^2}{2(1-\beta)^2} \|x_s - x_{s-1}\|^2 \leq 2\Delta_s^{(y)} + \frac{L\|m_{s-1}\|^2}{(1-\beta)^2},$$

where we apply Lemma A.1 and $\beta \in [0, 1)$. Finally dividing 2 on both sides and re-arranging the order, we obtain the desired result. $\qquad\square$

*Proof of Lemma 5.8.* Recalling the definition of $\boldsymbol{b}_s$ and $\boldsymbol{v}_s$, and then using Lemma A.3,

$$\sum_{s=1}^{t} \frac{g_{s,i}^2}{b_{s,i}^2} = \sum_{s=1}^{t} \frac{g_{s,i}^2}{(\sqrt{v_{s,i}} + \epsilon)^2} \le \sum_{s=1}^{t} \frac{g_{s,i}^2}{v_{s,i} + \epsilon^2} \le \log\left(1 + \frac{1}{\epsilon^2}\sum_{s=1}^{t} g_{s,i}^2\right). \tag{38}$$

Using the basic inequality, Assumption (A3), and Lemma A.1,

$$\sum_{j=1}^{t} g_{j,i}^2 \le \sum_{j=1}^{T} g_{j,i}^2 \le 2\sum_{j=1}^{T}(\bar{g}_{j,i}^2 + \xi_{j,i}^2) \le 2\sum_{j=1}^{T}(\|\bar{\boldsymbol{g}}_j\|^2 + \|\boldsymbol{\xi}_j\|^2)$$

$$\le 2A\sum_{j=1}^{T}\Delta_j^{(x)} + 2(B+1)\sum_{j=1}^{T}\|\bar{\boldsymbol{g}}_j\|^2 + 2CT \le X\sum_{j=1}^{T}\Delta_j^{(x)} + 2CT. \tag{39}$$

Combining with (38) and Lemma 5.2, we obtain that

$$\sum_{s=1}^{t} \frac{g_{s,i}^2}{b_{s,i}^2} \le \log\left(1 + \frac{1}{\epsilon^2}\sum_{j=1}^{T} g_{j,i}^2\right) \le \log\left(1 + \frac{1}{\epsilon^2}\left(X\sum_{j=1}^{T}\Delta_j^{(x)} + 2CT\right)\right) \le \log \mathcal{F}_T, \tag{40}$$

where $\mathcal{F}_T$ is defined as

$$\mathcal{F}_T := 1 + \frac{1}{\epsilon^2}\left[(\Delta_1^{(x)}X + 2C)T + \left(\frac{\eta\|\bar{\boldsymbol{g}}_1\|\sqrt{d}}{1-\beta} + \frac{L\eta^2 d}{2(1-\beta)^2}\right)XT^2 + \frac{L\eta^2 dXT^3}{(1-\beta)^2}\right]. \tag{41}$$

Summing up (40) over $i \in [d]$, we prove the first desired result.

Then we move to estimate terms related to $\boldsymbol{m}_s$. Let $M_s = \sum_{j=1}^{s}\beta^{s-j}$. For any $i \in [d]$, recalling (31) and using the convexity of the square function,

$$m_{s,i}^2 = \eta^2\left(\sum_{j=1}^{s}\frac{\beta^{s-j}g_{j,i}}{\sqrt{v_{j,i}} + \epsilon}\right)^2 \le \eta^2 M_s\sum_{j=1}^{s}\frac{\beta^{s-j}g_{j,i}^2}{(\sqrt{v_{j,i}} + \epsilon)^2} \le \eta^2 M_s\sum_{j=1}^{s}\frac{\beta^{s-j}g_{j,i}^2}{v_{j,i} + \epsilon^2}. \tag{42}$$

Summing over $i \in [d]$, using $\beta < 1$ with $M_s \le \frac{1}{1-\beta}$, Lemma A.3 and $s \le T$,

$$\|\boldsymbol{m}_s\|^2 \le \eta^2 M_s\sum_{i=1}^{d}\sum_{j=1}^{s}\frac{g_{j,i}^2}{\sqrt{\sum_{k=1}^{j} g_{k,i}^2} + \epsilon^2} \le \frac{\eta^2}{1-\beta}\sum_{i=1}^{d}\log\left(1 + \frac{1}{\epsilon^2}\sum_{j=1}^{s} g_{j,i}^2\right). \tag{43}$$

We also sum up (42) over $s \in [t]$ and apply Lemma A.3 to obtain that

$$\sum_{s=1}^{t} m_{s,i}^2 \le \eta^2\sum_{s=1}^{t} M_s\sum_{j=1}^{s}\frac{\beta^{s-j}g_{j,i}^2}{v_{j,i} + \epsilon^2} = \eta^2\sum_{j=1}^{t}\frac{g_{j,i}^2}{v_{j,i} + \epsilon^2}\sum_{s=j}^{t} M_s\beta^{s-j}$$

$$\le \frac{\eta^2}{(1-\beta)^2}\sum_{j=1}^{t}\frac{g_{j,i}^2}{v_{j,i} + \epsilon^2} \le \frac{\eta^2}{(1-\beta)^2}\log\left(1 + \frac{1}{\epsilon^2}\sum_{j=1}^{t} g_{j,i}^2\right).$$

Summing over $i \in [d]$, we obtain that

$$\sum_{s=1}^{t}\|\boldsymbol{m}_s\|^2 \le \frac{\eta^2}{(1-\beta)^2}\sum_{i=1}^{d}\log\left(1 + \frac{1}{\epsilon^2}\sum_{j=1}^{t} g_{j,i}^2\right). \tag{44}$$

Finally, we could follow the similar analysis for (39) and (40) to deduce that the terms inside the logarithm operator in (44) could be further bounded by $\mathcal{F}_T$ and thereby verify the target results. $\qquad\square$

## C    PROOF OF CONVERGENCE UNDER THE GENERALIZED SMOOTHNESS

In this section, we provide the detailed proof of Theorem 2. We still follow all the notations defined in Section 5. We shall first introduce two sequences $\{\mathcal{H}_s\}_{s\geq 1}$ and $\{\mathcal{L}_s\}_{s\geq 1}$ as follows:

$$\mathcal{H}_s = \sqrt{2A\Delta_s^{(x)} + 2(B+1)\left(4L_1\Delta_s^{(x)} + \sqrt{4L_0\Delta_s^{(x)}}\right)^2 + 2C}, \quad \mathcal{L}_s = 2L_0 + 2L_1\left(4L_1\Delta_s^{(x)} + \sqrt{4L_0\Delta_s^{(x)}}\right). \quad (45)$$

As a consequence, we slightly change the proxy step-size $\boldsymbol{a}_s$ in (12) as

$$\tilde{\boldsymbol{a}}_s = \sqrt{\boldsymbol{v}_{s-1} + (\mathcal{H}_s\boldsymbol{1}_d)^2} + \boldsymbol{\epsilon}, \quad \forall s \in [T]. \quad (46)$$

### C.1    CONVERGENCE OF ADAGRAD

As a consequence of Theorem 2, we obtain the following convergence bound for AdaGrad considering affine variance noise and the generalized smoothness.

**Corollary 2.** *Let $T \geq 1$ and $\delta \in (0,1)$. Suppose that $\{\boldsymbol{x}_s\}_{s\in[T]}$ is a sequence generated by Algorithm 1 with $\beta = 0$, $f$ is $(L_0, L_1)$-smooth satisfying (25), Assumptions (A1), (A2) hold and Assumption (A3) holds with $A = 0$, and the parameters follow the condition in (27) with $\beta = 0$, $\mathcal{H}, \mathcal{L}$ follow the definitions in (28),*

$$\Lambda_y \sim \mathcal{O}\left(\Delta_1^{(x)} + C_0^2 d \log\left(\frac{T}{\delta} + \frac{T}{\epsilon^2}\right)\right), \quad \Lambda_x \sim \mathcal{O}\left(\Lambda_y^2\right).$$

*Then it holds that with probability at least $1 - \delta$,*

$$\frac{1}{T}\sum_{s=1}^{T}\|\nabla f(\boldsymbol{x}_s)\|^2 \leq \frac{4\Lambda_y}{\eta}\left(\frac{2\Lambda_y(B+1)/\eta + \mathcal{H} + \epsilon}{T} + \sqrt{\frac{2C}{T}}\right).$$

**Remark C.1.** *Wang et al. [2023] provided a convergence rate for AdaGrad-Norm under the generalized smoothness with the expected affine variance noise, specifically when $\eta < \frac{1}{L_1}\min\left\{\frac{1}{64B}, \frac{1}{8\sqrt{B}}\right\}$, with probability at least $1 - \delta$,*

$$\min_{t\in[T]}\|\nabla f(\boldsymbol{x}_t)\|^2 = \mathcal{O}\left(\frac{\log(\sqrt{C}T)}{T\delta^2} + \frac{\sqrt{C}\log(\sqrt{C}T)}{\sqrt{T}\delta^2}\right). \quad (47)$$

*Thus, our convergence bound in Corollary 2 could reduce to the AdaGrad-Norm case and match the rate in (47) up to logarithm factors, while with a better dependency on the probability margin $\delta$.*

### C.2    TECHNICAL LEMMAS

We provide an equivalent result in [Zhang et al., 2020a, Lemma A.2], which establishes a different relationship of the gradient norm and the function value gap. We refer to the proof of [Zhang et al., 2020a, Lemma A.2].

**Lemma C.1.** *Suppose that $f$ is $(L_0, L_1)$-smooth and Assumption (A1) holds. Then, for any $\boldsymbol{x} \in \mathbb{R}^d$,*

$$\|\nabla f(\boldsymbol{x})\| \leq \max\left\{4L_1(f(\boldsymbol{x}) - f^*), \sqrt{4L_0(f(\boldsymbol{x}) - f^*)}\right\}.$$

We also have the following lemma to ensure the distance of $\boldsymbol{y}_s$ and $\boldsymbol{x}_s$ within $1/L_1$ in order to ensure the generalized smoothness in (25).

**Lemma C.2.** *Let $\boldsymbol{x}_s, \boldsymbol{y}_s$ be as in Algorithm 1 and (7). If $\beta \in [0,1)$, then for any $s \geq 1$,*

$$\max\{\|\boldsymbol{x}_{s+1} - \boldsymbol{x}_s\|, \|\boldsymbol{y}_s - \boldsymbol{x}_s\|, \|\boldsymbol{y}_{s+1} - \boldsymbol{y}_s\|\} \leq \frac{\eta\sqrt{d}}{(1-\beta)^2}. \quad (48)$$

*As a consequence, when*

$$\eta \leq \frac{(1-\beta)^2}{L_1\sqrt{d}}, \quad then, \quad \max\{\|\boldsymbol{x}_{s+1} - \boldsymbol{x}_s\|, \|\boldsymbol{y}_s - \boldsymbol{x}_s\|, \|\boldsymbol{y}_{s+1} - \boldsymbol{y}_s\|\} \leq \frac{1}{L_1}, \quad \forall s \geq 1. \quad (49)$$

*Proof.* Recalling in Lemma 5.1, we have already bounded $\|\boldsymbol{m}_s\|$ that is independent from smooth-related conditions as follows:

$$\|\boldsymbol{x}_{s+1} - \boldsymbol{x}_s\| = \|\boldsymbol{m}_s\| \leq \frac{\eta\sqrt{d}}{1-\beta}, \quad \forall s \geq 1. \tag{50}$$

Applying the definition of $\boldsymbol{y}_s$ in (7), (50) and $\beta \in [0, 1)$,[1]

$$\|\boldsymbol{y}_s - \boldsymbol{x}_s\| = \frac{\beta}{1-\beta}\|\boldsymbol{x}_s - \boldsymbol{x}_{s-1}\| \leq \frac{\eta\sqrt{d}}{(1-\beta)^2}, \quad \forall s \geq 1. \tag{51}$$

Using the iteration of $\boldsymbol{y}_s$ in (8) and Young's inequality

$$\|\boldsymbol{y}_{s+1} - \boldsymbol{y}_s\| = \frac{\eta}{1-\beta}\left\|\frac{\boldsymbol{g}_s}{\boldsymbol{b}_s}\right\| \leq \frac{\eta\sqrt{d}}{1-\beta}\left\|\frac{\boldsymbol{g}_s}{\boldsymbol{b}_s}\right\|_\infty \leq \frac{\eta\sqrt{d}}{1-\beta}, \quad \forall s \geq 1, \tag{52}$$

where we apply $|g_{s,i}/b_{s,i}| \leq 1, \forall i \in [d]$ in the last inequality. Combining with (50), (51) and (52), and using $\beta \in [0, 1)$, we then deduce an uniform bound for all three gaps. Finally, letting $\frac{\eta\sqrt{d}}{(1-\beta)^2} \leq \frac{1}{L_1}$, we then prove that (49) holds. $\square$

**Lemma C.3.** *Suppose that (49) holds. If $f$ is $(L_0, L_1)$-smooth, then for any $s \geq 1$,*

$$\begin{aligned}
\|\nabla f(\boldsymbol{x}_{s+1}) - \nabla f(\boldsymbol{x}_s)\| &\leq \mathcal{L}_s\|\boldsymbol{x}_{s+1} - \boldsymbol{x}_s\|, \\
\|\nabla f(\boldsymbol{y}_s) - \nabla f(\boldsymbol{x}_s)\| &\leq \mathcal{L}_s\|\boldsymbol{y}_s - \boldsymbol{x}_s\|, \\
\|\nabla f(\boldsymbol{y}_{s+1}) - \nabla f(\boldsymbol{y}_s)\| &\leq \mathcal{L}_s\|\boldsymbol{y}_{s+1} - \boldsymbol{y}_s\|.
\end{aligned} \tag{53}$$

*As a consequence, for any $s \geq 1$,*

$$\begin{aligned}
f(\boldsymbol{y}_{s+1}) - f(\boldsymbol{y}_s) - \langle\nabla f(\boldsymbol{y}_s), \boldsymbol{y}_{s+1} - \boldsymbol{y}_s\rangle &\leq \frac{\mathcal{L}_s}{2}\|\boldsymbol{y}_{s+1} - \boldsymbol{y}_s\|^2, \\
f(\boldsymbol{x}_s) - f(\boldsymbol{y}_s) - \langle\nabla f(\boldsymbol{y}_s), \boldsymbol{x}_s - \boldsymbol{y}_s\rangle &\leq \frac{\mathcal{L}_s}{2}\|\boldsymbol{x}_s - \boldsymbol{y}_s\|^2, \\
f(\boldsymbol{x}_{s+1}) - f(\boldsymbol{x}_s) - \langle\nabla f(\boldsymbol{x}_s), \boldsymbol{x}_{s+1} - \boldsymbol{x}_s\rangle &\leq \frac{\mathcal{L}_s}{2}\|\boldsymbol{x}_{s+1} - \boldsymbol{x}_s\|^2.
\end{aligned} \tag{54}$$

*Proof.* Noting that when (49) holds, we could use $\|\boldsymbol{y}_s - \boldsymbol{x}_s\| \leq 1/L_1$ and the generalized smoothness in (25) to get that

$$\begin{aligned}
\|\nabla f(\boldsymbol{y}_s)\| &\leq \|\nabla f(\boldsymbol{x}_s)\| + \|\nabla f(\boldsymbol{y}_s) - \nabla f(\boldsymbol{x}_s)\| \\
&\leq \|\nabla f(\boldsymbol{x}_s)\| + (L_0 + L_1\|\nabla f(\boldsymbol{x}_s)\|)\|\boldsymbol{y}_s - \boldsymbol{x}_s\| \leq 2\|\nabla f(\boldsymbol{x}_s)\| + L_0/L_1.
\end{aligned}$$

Using Lemma C.1 and combining with $\mathcal{L}_s$ in (45), we have

$$L_0 + L_1\|\nabla f(\boldsymbol{x}_s)\| \leq L_0 + L_1\left(4L_1\Delta_s^{(x)} + \sqrt{4L_0\Delta_s^{(x)}}\right) \leq \mathcal{L}_s,$$

$$L_0 + L_1\|\nabla f(\boldsymbol{y}_s)\| \leq 2L_0 + 2L_1\|\nabla f(\boldsymbol{x}_s)\| \leq 2L_0 + 2L_1\left(4L_1\Delta_s^{(x)} + \sqrt{4L_0\Delta_s^{(x)}}\right) \leq \mathcal{L}_s. \tag{55}$$

Thus, combining with (25), we prove (53). Now based on (53), we could deduce (54). We refer to the proof of [Zhang et al., 2020a, Lemma A.3]. $\square$

## C.3 ROUGH ESTIMATIONS

In generalized smooth cases, we revise the estimations in Lemmas 5.1 and 5.2 as follows.

**Lemma C.4.** *Suppose that $f$ is $(L_0, L_1)$-smooth, $\beta \in [0, 1)$ and (49) holds. Then, for any $s \geq 1$,*

$$\|\boldsymbol{m}_s\| \leq \frac{\eta\sqrt{d}}{1-\beta}, \quad \|\bar{\boldsymbol{g}}_s\| \leq \|\bar{\boldsymbol{g}}_1\| + \frac{\eta\sqrt{d}}{1-\beta}\sum_{j=1}^{s}\mathcal{L}_j.$$

---

[1]The inequality still holds for $s = 1$ since $\boldsymbol{x}_1 = \boldsymbol{y}_1$.

*Proof.* First, the estimation for $\|\boldsymbol{m}_s\|$ remains unchanged as in Lemma 5.1 since it does not rely on smooth-related conditions. Note that (49) holds. Then using (53), for any $s \geq 2$,

$$\|\bar{\boldsymbol{g}}_s\| \leq \|\bar{\boldsymbol{g}}_{s-1}\| + \|\bar{\boldsymbol{g}}_s - \bar{\boldsymbol{g}}_{s-1}\| \leq \|\bar{\boldsymbol{g}}_{s-1}\| + \mathcal{L}_{s-1}\|\boldsymbol{x}_s - \boldsymbol{x}_{s-1}\| = \|\bar{\boldsymbol{g}}_{s-1}\| + \mathcal{L}_{s-1}\|\boldsymbol{m}_{s-1}\|. \tag{56}$$

Further using (56) and the estimation for $\|\boldsymbol{m}_s\|$, for any $s \geq 2$,

$$\|\bar{\boldsymbol{g}}_s\| \leq \|\bar{\boldsymbol{g}}_{s-1}\| + \frac{\mathcal{L}_{s-1}\eta\sqrt{d}}{1-\beta} \leq \|\bar{\boldsymbol{g}}_1\| + \frac{\eta\sqrt{d}}{1-\beta}\sum_{j=1}^{s-1}\mathcal{L}_j.$$

Note that the above inequality also holds when $s = 1$. Thus, the proof is complete. $\quad\square$

**Lemma C.5.** *Under the same conditions of Lemma C.4, for any $T \geq 1$,*

$$\sum_{t=1}^{T}\Delta_t^{(x)} \leq \mathcal{I}_T, \quad \mathcal{I}_T := \Delta_1^{(x)}T + \frac{\eta\sqrt{d}}{1-\beta}\sum_{t=1}^{T}\sum_{s=1}^{t}\left(\|\bar{\boldsymbol{g}}_1\| + \frac{\eta\sqrt{d}}{1-\beta}\sum_{j=1}^{s}\mathcal{L}_j\right) + \frac{\eta^2 d}{2(1-\beta)^2}\sum_{t=1}^{T}\sum_{s=1}^{t}\mathcal{L}_s. \tag{57}$$

*Proof.* Since (49) holds, we could rely on the updated rule in Algorithm 1 and (54) to obtain that

$$f(\boldsymbol{x}_{s+1}) \leq f(\boldsymbol{x}_s) + \langle\bar{\boldsymbol{g}}_s, \boldsymbol{x}_{s+1} - \boldsymbol{x}_s\rangle + \frac{\mathcal{L}_s}{2}\|\boldsymbol{x}_{s+1} - \boldsymbol{x}_s\|^2 = f(\boldsymbol{x}_s) + \langle\bar{\boldsymbol{g}}_s, \boldsymbol{m}_s\rangle + \frac{\mathcal{L}_s}{2}\|\boldsymbol{m}_s\|^2. \tag{58}$$

Using Cauchy-Schwarz inequality and Lemma C.4, for any $s \geq 1$,

$$\langle\bar{\boldsymbol{g}}_s, \boldsymbol{m}_s\rangle \leq \|\bar{\boldsymbol{g}}_s\|\|\boldsymbol{m}_s\| \leq \frac{\eta\sqrt{d}}{1-\beta}\left(\|\bar{\boldsymbol{g}}_1\| + \frac{\eta\sqrt{d}}{1-\beta}\sum_{j=1}^{s}\mathcal{L}_j\right), \quad \frac{\mathcal{L}_s}{2}\|\boldsymbol{m}_s\|^2 \leq \frac{\mathcal{L}_s\eta^2 d}{2(1-\beta)^2}.$$

Combining the above, subtracting $f^*$ on both sides of (58) and summing up over $s \in [t]$, we obtain that for any $t \geq 1$,

$$\Delta_{t+1}^{(x)} \leq \Delta_1^{(x)} + \frac{\eta\sqrt{d}}{1-\beta}\sum_{s=1}^{t}\left(\|\bar{\boldsymbol{g}}_1\| + \frac{\eta\sqrt{d}}{1-\beta}\sum_{j=1}^{s}\mathcal{L}_j\right) + \frac{\eta^2 d}{2(1-\beta)^2}\sum_{s=1}^{t}\mathcal{L}_s.$$

We define $\sum_a^b = 0$ when $a < b$. Then, we sum up over $t \in [0, 1, \cdots T - 1]$ and obtain the desired result. $\quad\square$

## C.4   START POINT AND DECOMPOSITION

To start with, we recall $\eta$ in (27) and verify that (49) always holds. Then, we could use the descent lemma (54) and apply (8), and sum up both sides over $s \in [t]$ to get that

$$\begin{aligned}
f(\boldsymbol{y}_{t+1}) &\leq f(\boldsymbol{x}_1) + \sum_{s=1}^{t}\langle\nabla f(\boldsymbol{y}_s), \boldsymbol{y}_{s+1} - \boldsymbol{y}_s\rangle + \sum_{s=1}^{t}\frac{\mathcal{L}_s}{2}\|\boldsymbol{y}_{s+1} - \boldsymbol{y}_s\|^2 \\
&= f(\boldsymbol{x}_1) + \frac{\eta}{1-\beta}\cdot\mathbf{A} + \frac{\eta^2}{2(1-\beta)^2}\sum_{s=1}^{t}\mathcal{L}_s\left\|\frac{\boldsymbol{g}_s}{\boldsymbol{b}_s}\right\|^2,
\end{aligned} \tag{59}$$

where we use $\boldsymbol{x}_1 = \boldsymbol{y}_1$ and the definition of $\mathbf{A}$ in (10). We follow the decomposition in (60) and restate as follows,

$$\mathbf{A} = \underbrace{-\sum_{s=1}^{t}\left\langle\bar{\boldsymbol{g}}_s, \frac{\boldsymbol{g}_s}{\boldsymbol{b}_s}\right\rangle}_{\mathbf{A.1}} + \underbrace{\sum_{s=1}^{t}\left\langle\bar{\boldsymbol{g}}_s - \nabla f(\boldsymbol{y}_s), \frac{\boldsymbol{g}_s}{\boldsymbol{b}_s}\right\rangle}_{\mathbf{A.2}}. \tag{60}$$

## C.5 ESTIMATING A

We then introduce $\tilde{\boldsymbol{a}}$ in (46) into (60) to derive that

$$\mathbf{A.1} = -\sum_{s=1}^{t}\left\|\frac{\bar{\boldsymbol{g}}_s}{\sqrt{\tilde{\boldsymbol{a}}_s}}\right\|^2 \underbrace{-\sum_{s=1}^{t}\left\langle \bar{\boldsymbol{g}}_s, \frac{\boldsymbol{\xi}_s}{\tilde{\boldsymbol{a}}_s}\right\rangle}_{\mathbf{A'.1.1}} + \underbrace{\sum_{s=1}^{t}\left\langle \bar{\boldsymbol{g}}_s, \left(\frac{1}{\tilde{\boldsymbol{a}}_s} - \frac{1}{\boldsymbol{b}_s}\right)\boldsymbol{g}_s\right\rangle}_{\mathbf{A'.1.2}}. \tag{61}$$

The technique for estimating $\mathbf{A'.1.1}$ is similar to Lemma 5.3. Let $X_s' = -\left\langle \bar{\boldsymbol{g}}_s, \frac{\boldsymbol{\xi}_s}{\tilde{\boldsymbol{a}}_s}\right\rangle$. We could still verify that $X_s'$ is a martingale difference sequence and define $\zeta_s' = \left\|\frac{\bar{\boldsymbol{g}}_s}{\tilde{\boldsymbol{a}}_s}\right\| \sqrt{A\Delta_s^{(x)} + B\|\bar{\boldsymbol{g}}_s\|^2 + C}$. Similarly, $\zeta_s'$ is a random variable dependent on $\boldsymbol{z}_1, \cdots, \boldsymbol{z}_{s-1}$. Using Cauchy-Schwarz inequality and Assumption (A3), we have

$$\mathbb{E}\left[\exp\left[\left(\frac{X_s'}{\zeta_s'}\right)^2\right] \mid \boldsymbol{z}_1, \cdots, \boldsymbol{z}_{s-1}\right] \leq \mathbb{E}\left[\exp\left(\frac{\|\boldsymbol{\xi}_s\|^2}{A\Delta_s^{(x)} + B\|\bar{\boldsymbol{g}}_s\|^2 + C}\right) \mid \boldsymbol{z}_1, \cdots, \boldsymbol{z}_{s-1}\right] \leq \mathrm{e}.$$

However, using Lemma C.1 and $\mathcal{H}_s$ in (45), we derive that

$$A\Delta_s^{(x)} + B\|\bar{\boldsymbol{g}}_s\|^2 + C \leq A\Delta_s^{(x)} + B\left(4L_1\Delta_s^{(x)} + \sqrt{4L_0\Delta_s^{(x)}}\right)^2 + C \leq \mathcal{H}_s^2.$$

Hence, we obtain a different inequality from (34) that

$$\mathbf{A'.1.1} \leq \frac{3\lambda}{4}\sum_{s=1}^{t}\sum_{i=1}^{d}\frac{\bar{g}_{s,i}^2}{\tilde{a}_{s,i}^2}\left(A\Delta_s^{(x)} + B\|\bar{\boldsymbol{g}}_s\|^2 + C\right) + \frac{1}{\lambda}\log\left(\frac{1}{\delta}\right)$$

$$\leq \frac{3\lambda}{4}\sum_{s=1}^{t}\sum_{i=1}^{d}\frac{\bar{g}_{s,i}^2\mathcal{H}_s^2}{\tilde{a}_{s,i}^2} + \frac{1}{\lambda}\log\left(\frac{1}{\delta}\right) \leq \frac{3\lambda}{4}\sum_{s=1}^{t}\sum_{i=1}^{d}\frac{\bar{g}_{s,i}^2\mathcal{H}_s}{\tilde{a}_{s,i}} + \frac{1}{\lambda}\log\left(\frac{1}{\delta}\right), \tag{62}$$

where the last inequality applies $1/\tilde{a}_{s,i} \leq 1/\mathcal{H}_s$ from (46). Then, we can re-scale $\delta$ and take $\lambda = 1/(3\mathcal{H})$, leading to the new estimation as follows: with probability at least $1 - \delta$,

$$\mathbf{A'.1.1} \leq \frac{1}{4}\sum_{s=1}^{t}\frac{\mathcal{H}_s}{\mathcal{H}}\left\|\frac{\bar{\boldsymbol{g}}_s}{\sqrt{\tilde{\boldsymbol{a}}_s}}\right\|^2 + 3\mathcal{H}\log\left(\frac{T}{\delta}\right), \quad \forall t \in [T]. \tag{63}$$

The estimation for $\mathbf{A'.1.2}$ remains similar to (15). We first derive from the basic inequality and Assumption (A3) that

$$\|\boldsymbol{g}_s\|^2 \leq 2\|\bar{\boldsymbol{g}}_s\|^2 + 2\|\boldsymbol{\xi}_s\|^2 \leq 2A\Delta_s^{(x)} + 2(B+1)\|\bar{\boldsymbol{g}}_s\|^2 + 2C \leq \mathcal{H}_s^2.$$

Then, based on $\|\boldsymbol{g}_s\|^2 \leq \mathcal{H}_s^2$, we derive a similar result as in (36) where

$$\left|\frac{1}{\tilde{a}_{s,i}} - \frac{1}{b_{s,i}}\right| \leq \frac{\mathcal{H}_s}{\tilde{a}_{s,i}b_{s,i}}, \quad \forall s \in [T], \forall i \in [d].$$

Then, using a similar deduction in (15), we derive that

$$\mathbf{A'.1.2} \leq \frac{1}{4}\sum_{s=1}^{t}\left\|\frac{\bar{\boldsymbol{g}}_s}{\sqrt{\tilde{\boldsymbol{a}}_s}}\right\|^2 + \sum_{s=1}^{t}\mathcal{H}_s\left\|\frac{\boldsymbol{g}_s}{\boldsymbol{b}_s}\right\|^2. \tag{64}$$

The estimation for $\mathbf{A.2}$ in (60) is revised by the following lemma.

**Lemma C.6.** *Suppose that $f$ is $(L_0, L_1)$-smooth and (49) holds. For any $t \geq 1$, if $\beta \in [0, 1)$, it holds that*

$$\mathbf{A.2} \leq \sum_{s=1}^{t}\frac{\mathcal{L}_s}{2\eta}\|\boldsymbol{m}_{s-1}\|^2 + \sum_{s=1}^{t}\frac{\mathcal{L}_s\eta}{2(1-\beta)^2}\left\|\frac{\boldsymbol{g}_s}{\boldsymbol{b}_s}\right\|^2. \tag{65}$$

*Proof.* Noting that when (49) holds, we could rely on (53) and $\beta \in [0, 1)$ to obtain that

$$\|\bar{\boldsymbol{g}}_s - \nabla f(\boldsymbol{y}_s)\| \leq \mathcal{L}_s \|\boldsymbol{y}_s - \boldsymbol{x}_s\| = \frac{\mathcal{L}_s \beta}{1 - \beta} \|\boldsymbol{x}_s - \boldsymbol{x}_{s-1}\| \leq \frac{\mathcal{L}_s}{1 - \beta} \|\boldsymbol{m}_{s-1}\|. \tag{66}$$

Applying Cauchy-Schwarz inequality and using (66),

$$\mathbf{A.2} \leq \sum_{s=1}^{t} \|\bar{\boldsymbol{g}}_s - \nabla f(\boldsymbol{y}_s)\| \left\|\frac{\boldsymbol{g}_s}{\boldsymbol{b}_s}\right\| \leq \sum_{s=1}^{t} \frac{\mathcal{L}_s}{1 - \beta} \|\boldsymbol{m}_{s-1}\| \left\|\frac{\boldsymbol{g}_s}{\boldsymbol{b}_s}\right\| \leq \sum_{s=1}^{t} \frac{\mathcal{L}_s}{2\eta} \|\boldsymbol{m}_{s-1}\|^2 + \sum_{s=1}^{t} \frac{\mathcal{L}_s \eta}{2(1 - \beta)^2} \left\|\frac{\boldsymbol{g}_s}{\boldsymbol{b}_s}\right\|^2.$$

$\square$

## C.6 BOUNDING THE FUNCTION VALUE GAP

Based on the above estimations, we are now ready to provide the bound for the function value gap along the optimization trajectory in the following proposition.

**Proposition C.1.** *Under the same conditions in Theorem 2, for any given $\delta \in (0, 1)$, the following two inequalities hold with probability at least $1 - \delta$,*

$$\Delta_t^{(x)} \leq \Lambda_x, \quad \mathcal{H}_t \leq \mathcal{H}, \quad \mathcal{L}_t \leq \mathcal{L}, \quad \forall t \in [T + 1], \tag{67}$$

*and*

$$\Delta_{t+1}^{(y)} \leq \Lambda_y - \frac{\eta}{2(1 - \beta)} \sum_{s=1}^{t} \left\|\frac{\bar{\boldsymbol{g}}_s}{\sqrt{\tilde{\boldsymbol{a}}_s}}\right\|^2, \quad \forall t \in [T], \tag{68}$$

*where $\mathcal{H}_t, \mathcal{L}_t$ are as in (45) and $\Lambda_x, \Lambda_y, \mathcal{H}, \mathcal{L}$ are as in Theorem 2.*

In what follows, we prove Proposition C.1. First, we verify that (49) holds from the setting of $\eta$ in (27). Hence, we ensure that (59) and (65) hold.

In the following, we will assume the inequality (63) holds and then deduce the results in (67) and (68). Noting that (63) holds with probability at least $1 - \delta$, we thereby deduce that the desired results hold with probability at least $1 - \delta$.

Plugging the estimations (63) and (64) into (61), we obtain the bound for **A.1**. Then, combining with (65), (60) and (59), and subtracting $f^*$ on both sides of (59),

$$\Delta_{t+1}^{(y)} \leq \Delta_1^{(x)} + \frac{\eta}{1 - \beta} \sum_{s=1}^{t} \left(\frac{\mathcal{H}_s}{4\mathcal{H}} - \frac{3}{4}\right) \left\|\frac{\bar{\boldsymbol{g}}_s}{\sqrt{\tilde{\boldsymbol{a}}_s}}\right\|^2 + \frac{3\mathcal{H}\eta}{1 - \beta} \log\left(\frac{T}{\delta}\right) + \frac{\eta}{1 - \beta} \sum_{s=1}^{t} \mathcal{H}_s \left\|\frac{\boldsymbol{g}_s}{\boldsymbol{b}_s}\right\|^2$$

$$+ \sum_{s=1}^{t} \frac{\mathcal{L}_s}{2(1 - \beta)} \|\boldsymbol{m}_{s-1}\|^2 + \sum_{s=1}^{t} \left(\frac{\mathcal{L}_s \eta^2}{2(1 - \beta)^3} + \frac{\mathcal{L}_s \eta^2}{2(1 - \beta)^2}\right) \left\|\frac{\boldsymbol{g}_s}{\boldsymbol{b}_s}\right\|^2. \tag{69}$$

We still rely on an induction argument to deduce the result. First, we define two polynomials $\tilde{\mathcal{I}}_t$ and $\tilde{\mathcal{J}}_t$ with respect to $t$ and present the detailed expressions of $\Lambda_y$ and $\Lambda_x$ that are independent from $t$ as follows:

$$\tilde{\mathcal{I}}_t := \Delta_1^{(x)} \cdot t + \frac{C_0 \sqrt{d}}{1 - \beta} \left(\|\bar{\boldsymbol{g}}_1\| + \frac{C_0 \sqrt{d} t}{1 - \beta}\right) \cdot t^2 + \frac{C_0^2 d}{2(1 - \beta)^2} \cdot t^2, \tag{70}$$

$$\tilde{\mathcal{J}}_t := 1 + \frac{1}{\epsilon^2} \left[2A\tilde{\mathcal{I}}_t + 2(B + 1)\left(\|\bar{\boldsymbol{g}}_1\| + \frac{C_0 \sqrt{d} t}{1 - \beta}\right)^2 \cdot t + 2Ct\right], \tag{71}$$

$$\Lambda_y := \Delta_1^{(x)} + \frac{3C_0}{1 - \beta} \log\left(\frac{T}{\delta}\right) + \frac{C_0 d}{1 - \beta} \log \tilde{\mathcal{J}}_T,$$

$$+ \frac{C_0^2 d}{2(1 - \beta)^3} \log \tilde{\mathcal{J}}_T + \left(\frac{C_0^2 d}{2(1 - \beta)^3} + \frac{C_0^2 d}{2(1 - \beta)^2}\right) \log \tilde{\mathcal{J}}_T, \tag{72}$$

$$\Lambda_x := (2L_0 + 1)\Lambda_y + 8L_1 \Lambda_y^2 + \frac{C_0^2 d}{(1 - \beta)^2} \log \tilde{\mathcal{J}}_T. \tag{73}$$

It's worthy noting that $\tilde{\mathcal{I}}_t$ and $\tilde{\mathcal{J}}_t$ are deterministic polynomials with respect to $t$ and are dependent on hyper-parameters $C_0, \beta, d$ and problem-parameters $A, B, C$. We could verify that $\Lambda_y \sim \mathcal{O}(\log(T/\delta))$ and $\Lambda_x \sim \mathcal{O}(\log^2(T/\delta))$.

**Induction argument** The induction begins by noting that $\Delta_1^{(x)} \le \Lambda_x$ from (72) and (73). Suppose that for some $t \in [T]$,

$$\Delta_s^{(x)} \le \Lambda_x, \quad \text{thus,} \quad \mathcal{H}_s \le \mathcal{H}, \quad \mathcal{L}_s \le \mathcal{L}, \quad \forall s \in [t], \tag{74}$$

where we rely on $\mathcal{H}_s, \mathcal{L}_s$ in (45), and $\mathcal{H}, \mathcal{L}$ in (28). We thus apply (74) to (69) and get

$$\Delta_{t+1}^{(y)} \le \Delta_1^{(x)} - \frac{\eta}{2(1-\beta)} \sum_{s=1}^{t} \left\| \frac{\bar{\boldsymbol{g}}_s}{\sqrt{\tilde{\boldsymbol{a}}_s}} \right\|^2 + \frac{3\mathcal{H}\eta}{1-\beta} \log\left(\frac{T}{\delta}\right) + \frac{\eta\mathcal{H}}{1-\beta} \sum_{s=1}^{t} \left\| \frac{\boldsymbol{g}_s}{\boldsymbol{b}_s} \right\|^2$$

$$+ \frac{\mathcal{L}}{2(1-\beta)} \sum_{s=1}^{t} \|\boldsymbol{m}_{s-1}\|^2 + \left( \frac{\mathcal{L}\eta^2}{2(1-\beta)^3} + \frac{\mathcal{L}\eta^2}{2(1-\beta)^2} \right) \sum_{s=1}^{t} \left\| \frac{\boldsymbol{g}_s}{\boldsymbol{b}_s} \right\|^2. \tag{75}$$

Then, Lemmas 5.8 should be further revised under the generalized smooth condition as follows.

**Lemma C.7.** *Suppose that $f$ is $(L_0, L_1)$-smooth and (49) holds. Then for any $t \ge 1$,*

$$\sum_{s=1}^{t} \left\| \frac{\boldsymbol{g}_s}{\boldsymbol{b}_s} \right\|^2 \le d \log \mathcal{J}_t, \quad \|\boldsymbol{m}_t\|^2 \le \frac{\eta^2 d}{1-\beta} \log \mathcal{J}_t, \quad \sum_{s=1}^{t} \|\boldsymbol{m}_s\|^2 \le \frac{\eta^2 d}{(1-\beta)^2} \log \mathcal{J}_t,$$

*where $\mathcal{J}_t$ is a polynomial with respect to $t$ with the detailed expression in (76).*

*Proof.* Recall that the three estimations in (38), (43) and (44) remain unchanged as they are not dependent by smooth-related conditions. However, we shall revise the estimation for the term inside the logarithm operator. We also start from the basic inequality and Assumption (A3),

$$\sum_{j=1}^{t} g_{j,i}^2 \le 2 \sum_{j=1}^{t} (\bar{g}_{j,i}^2 + \xi_{j,i}^2) \le 2 \sum_{j=1}^{t} (\|\bar{\boldsymbol{g}}_j\|^2 + \|\boldsymbol{\xi}_j\|^2) \le 2 \sum_{j=1}^{t} (A\Delta_j^{(x)} + (B+1)\|\bar{\boldsymbol{g}}_j\|^2 + C).$$

Then, applying Lemma C.4 and Lemma C.5,

$$1 + \frac{1}{\epsilon^2} \sum_{j=1}^{t} g_{j,i}^2 \le \mathcal{J}_t := 1 + \frac{1}{\epsilon^2} \left[ 2A\mathcal{I}_t + 2(B+1) \sum_{s=1}^{t} \left( \|\bar{\boldsymbol{g}}_1\| + \frac{\eta\sqrt{d}}{1-\beta} \sum_{j=1}^{s} \mathcal{L}_j \right)^2 + 2Ct \right]. \tag{76}$$

Plugging (76) into (38), (43) and (44), we thereby deduce the final result. $\square$

Therefore, applying Lemma C.7 to (75), we obtain that

$$\Delta_{t+1}^{(y)} \le \Delta_1^{(x)} - \frac{\eta}{2(1-\beta)} \sum_{s=1}^{t} \left\| \frac{\bar{\boldsymbol{g}}_s}{\sqrt{\tilde{\boldsymbol{a}}_s}} \right\|^2 + \frac{3\mathcal{H}\eta}{1-\beta} \log\left(\frac{T}{\delta}\right) + \frac{\mathcal{H}\eta d}{1-\beta} \log \mathcal{J}_t$$

$$+ \frac{\mathcal{L}\eta^2 d}{2(1-\beta)^3} \log \mathcal{J}_t + \left( \frac{\mathcal{L}\eta^2 d}{2(1-\beta)^3} + \frac{\mathcal{L}\eta^2 d}{2(1-\beta)^2} \right) \log \mathcal{J}_t. \tag{77}$$

Using (27), we have that

$$\mathcal{H}\eta \le C_0, \quad \mathcal{L}\eta \le C_0, \quad \mathcal{L}\eta^2 \le C_0^2. \tag{78}$$

We should also note that $\mathcal{I}_t$ in (57) and $\mathcal{J}_t$ in (76) both include $\mathcal{L}_s, s \le t$. Then recalling $\tilde{\mathcal{I}}_t$ and $\tilde{\mathcal{J}}_t$ in (70) and (71), and applying $\mathcal{L}_s \le \mathcal{L}, \forall s \le t$ in (74) and (78), we have that for any $t \in [T]$,

$$\mathcal{I}_t \le \tilde{\mathcal{I}}_t \le \tilde{\mathcal{I}}_T, \quad \mathcal{J}_t \le \tilde{\mathcal{J}}_t \le \tilde{\mathcal{J}}_T, \quad \log \mathcal{J}_t \le \log \tilde{\mathcal{J}}_t \le \log \tilde{\mathcal{J}}_T. \tag{79}$$

Then, applying (78) and (79) to (77), and recalling $\Lambda_y$ in (72),

$$\Delta_{t+1}^{(y)} \le \Lambda_y - \frac{\eta}{2(1-\beta)} \sum_{s=1}^{t} \left\| \frac{\bar{\boldsymbol{g}}_s}{\sqrt{\tilde{\boldsymbol{a}}_s}} \right\|^2 \le \Lambda_y. \tag{80}$$

Finally, we use the following lemma to control $\Delta_s^{(x)}$ by $\Delta_s^{(y)}$.

**Lemma C.8.** *Suppose that $f$ is $(L_0, L_q)$-smooth and (49) holds. Let $\boldsymbol{y}_s$ be defined in (7) and $\beta \in [0,1)$. Then for any $s \geq 1$,*

$$\Delta_s^{(x)} \leq (2L_0 + 1)\Delta_s^{(y)} + 8L_1 \left(\Delta_s^{(y)}\right)^2 + \frac{\mathcal{L}_s + 1}{2(1-\beta)^2}\|\boldsymbol{m}_{s-1}\|^2.$$

*Proof.* Since (49) holds, then using (54),

$$f(\boldsymbol{x}_s) \leq f(\boldsymbol{y}_s) + \langle \nabla f(\boldsymbol{y}_s), \boldsymbol{x}_s - \boldsymbol{y}_s \rangle + \frac{\mathcal{L}_s}{2}\|\boldsymbol{x}_s - \boldsymbol{y}_s\|^2$$

$$= f(\boldsymbol{y}_s) - \frac{\beta}{1-\beta}\langle \nabla f(\boldsymbol{y}_s), \boldsymbol{x}_s - \boldsymbol{x}_{s-1} \rangle + \frac{\mathcal{L}_s \beta^2}{2(1-\beta)^2}\|\boldsymbol{x}_s - \boldsymbol{x}_{s-1}\|^2.$$

Using Young's inequality and subtracting $f^*$ on both sides,

$$\Delta_s^{(x)} \leq \Delta_s^{(y)} + \frac{1}{2}\|\nabla f(\boldsymbol{y}_s)\|^2 + \frac{(\mathcal{L}_s + 1)\beta^2}{2(1-\beta)^2}\|\boldsymbol{x}_s - \boldsymbol{x}_{s-1}\|^2 \leq \Delta_s^{(y)} + 8L_1\left(\Delta_s^{(y)}\right)^2 + 2L_0\Delta_s^{(y)} + \frac{\mathcal{L}_s + 1}{2(1-\beta)^2}\|\boldsymbol{m}_{s-1}\|^2,$$

where we apply Lemma C.1 and $\beta \in [0,1)$ for the last inequality. $\qquad\square$

Using Lemma C.8, (80), and Lemma C.7, we could bound $\Delta_{t+1}^{(x)}$ as

$$\Delta_{t+1}^{(x)} \leq (2L_0 + 1)\Lambda_y + 8L_1\Lambda_y^2 + \frac{(\mathcal{L}_s + 1)\eta^2 d}{2(1-\beta)^2}\log \mathcal{J}_t \leq (2L_0 + 1)\Lambda_y + 8L_1\Lambda_y^2 + \frac{2C_0^2 d}{2(1-\beta)^2}\log \tilde{\mathcal{J}}_T,$$

where the last inequality applies (78) and (79). Recalling $\Lambda_x$ in (73), we find that

$$\Delta_{t+1}^{(x)} \leq \Lambda_x.$$

Combining with (74), the induction is thus complete. We prove the desired result in (67). Finally, as an intermediate result, (68) is verified by (80). The proof of Proposition C.1 is complete.

## C.7 PROOF OF THE MAIN RESULT

Based on Proposition C.1, we could prove the final convergence result as follows.

*Proof of Theorem 2.* The proof for the final convergence rate follows the similar idea and some same estimations in the proof of Theorem 1. Setting $t = T$ in (68), it holds that with probability at least $1 - \delta$,

$$\frac{\eta}{2(1-\beta)}\sum_{s=1}^{T}\frac{\|\bar{\boldsymbol{g}}_s\|^2}{\|\tilde{\boldsymbol{a}}_s\|_\infty} \leq \frac{\eta}{2(1-\beta)}\sum_{s=1}^{T}\left\|\frac{\bar{\boldsymbol{g}}_s}{\sqrt{\tilde{\boldsymbol{a}}_s}}\right\|^2 \leq \Lambda_y. \tag{81}$$

In what follows, we assume that (67) and (68) always hold and deduce the convergence bound under these two inequalities. Note that (67) and (68) hold with probability at least $1 - \delta$ according to Proposition C.1. Thus, the final convergence bound also holds with probability at least $1 - \delta$. Applying $\tilde{\boldsymbol{a}}_s$ in (46) and (67), and following the similar analysis in (24),

$$\|\tilde{\boldsymbol{a}}_s\|_\infty - \epsilon \leq \sqrt{2\sum_{j=1}^{s-1}(A\Delta_j^{(x)} + (B+1)\|\bar{\boldsymbol{g}}_j\|^2 + C) + \mathcal{H}_s^2} \leq \sqrt{2(B+1)\sum_{j=1}^{s-1}\|\bar{\boldsymbol{g}}_j\|^2 + 2(A\Lambda_x + C)s + \mathcal{H}^2}$$

$$\leq \sqrt{2(B+1)\sum_{s=1}^{T}\|\bar{\boldsymbol{g}}_s\|^2 + 2(A\Lambda_x + C)T + \mathcal{H}^2}, \quad \forall s \in [T].$$

Then combining with (81), and letting $\tilde{\Lambda}_y = \frac{2\Lambda_y(1-\beta)}{\eta}$,

$$\sum_{s=1}^{T} \|\bar{\boldsymbol{g}}_s\|^2 \leq \tilde{\Lambda}_y \left( \sqrt{2(B+1)\sum_{s=1}^{T} \|\bar{\boldsymbol{g}}_s\|^2 + 2(A\Lambda_x + C)T + \mathcal{H}^2 + \epsilon} \right)$$

$$\leq \tilde{\Lambda}_y \left( \sqrt{2(B+1)\sum_{s=1}^{T} \|\bar{\boldsymbol{g}}_s\|^2} + \sqrt{2(A\Lambda_x + C)T} + \mathcal{H} + \epsilon \right)$$

$$\leq \sum_{s=1}^{T} \frac{\|\bar{\boldsymbol{g}}_s\|^2}{2} + \tilde{\Lambda}_y^2(B+1) + \tilde{\Lambda}_y \left( \sqrt{2(A\Lambda_x + C)T} + \mathcal{H} + \epsilon \right),$$

where we apply Young's inequality for the last inequality. Re-arranging the order and dividing $T$ on both sides, we get

$$\frac{1}{T}\sum_{s=1}^{T} \|\bar{\boldsymbol{g}}_s\|^2 \leq 2\tilde{\Lambda}_y \left( \frac{\tilde{\Lambda}_y(B+1) + \mathcal{H} + \epsilon}{T} + \sqrt{\frac{2(A\Lambda_x + C)}{T}} \right).$$

$\square$

## D   EXPERIMENT

In this section, we present an experiment to verify that AdaGrad can find a stationary point when the noise satisfies Assumption (A3).

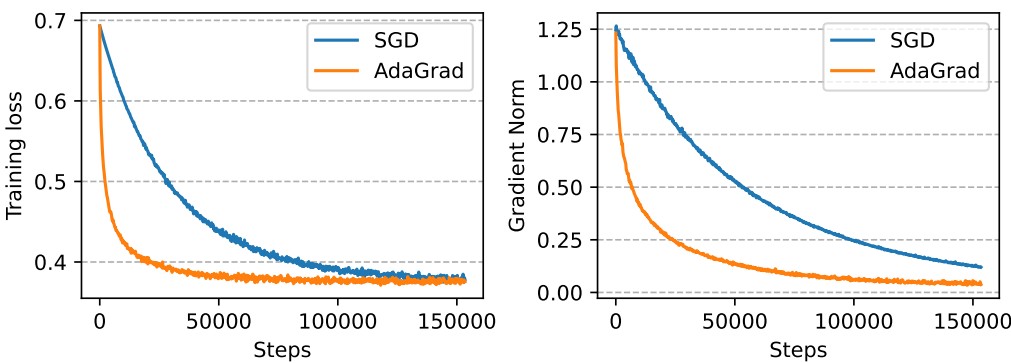

Figure 1: Training loss vs. steps and gradient norms vs. steps using SGD and AdaGrad.

**Experimental Setup**   We follow the experimental task outlined in [Khaled and Richtárik, 2023], where the objective is to minimize a regularized logistic regression problem defined as follows:

$$\min_{x \in \mathbb{R}^d} \left\{ \frac{1}{n}\sum_{i=1}^{n} \log\left(1 + \exp(-a_i^\top x)\right) + \lambda \sum_{j=1}^{d} \frac{x_j^2}{1 + x_j^2} \right\}. \tag{82}$$

In [Khaled and Richtárik, 2023], it was verified that using uniform sampling over the a9a dataset and the loss function in (82), the noise conforms to Assumption (A3) with $\hat{A} = 10.09, \hat{B} = 0, \hat{C} = 0.373$, which closely aligns with the theoretical values where $A = 9, B = 0, C = 0.994$. We then executed both SGD and AdaGrad to minimize (82) using a batch size of 256. The learning rates were set to $7 \times 10^{-6}$ for SGD and $5 \times 10^{-4}$ for AdaGrad.

We utilized the a9a dataset and the PyTorch implementations of SGD and AdaGrad. The experiments were conducted on a single NVIDIA GeForce RTX 4090 GPU.

**Results**   We plotted the training loss and gradient norms against the number of steps in Figure 1, training for 1200 epochs. The results indicate that both SGD and AdaGrad can find a stationary point given a sufficiently large number of steps $T$, thereby supporting the conclusion in Theorem 1.

