# OpenReview forum: "Revisiting Convergence of AdaGrad with Relaxed Assumptions"
_auai.org/UAI/2024/Conference — UAI 2024 poster_

### Official Review · Reviewer_JikW · 2024-02-29

**Q2-1 Originality-Novelty:** 3
**Q2-2 Correctness-Technical Quality:** 3
**Q2-5 Clarity Of Writing:** 4

**Q1 Summary And Contributions:**

This paper provides a probabilistic convergence analysis of AdaGrad with momentum on non-convex smooth optimization under a general noise assumption in (2).

**Q2-3 Extent To Which Claims Are Supported By Evidence:**

4: Excellent: all claims are supported by very convincing evidence (in the form of comprehensive experimental evaluation, rigorous mathematical proofs, detailed (pseudo-)code, precise references, well-motivated and realistic assumptions) and the authors deliver what they promise.

**Q2-4 Reproducibility:**

4: Excellent: key resources (e.g. proofs, code, data) are available and key details (e.g. proof sketches, experimental setup) are comprehensively described for competent researchers to confidently and easily reproduce the main results.

**Q3 Main Strengths:**

- The paper is well-written. The authors successfully present their contributions in comparison with many related references.
- The paper establishes new convergence results of AdaGrad with momentum under a broader assumption.

**Q4 Main Weakness:**

- The difference between the paper and previous work mainly relies on the (slightly generalized) assumptions. Hence, the contributions may not be groundbreaking.

**Q5 Detailed Comments To The Authors:**

NA.

**Q9 Complying With Reviewing Instructions:**

Yes

---

> ### Author Rebuttal · Authors · 2024-04-07
>
> We appreciate for your effort to review our manuscript and provide positive and valuable suggestions.
>
> ---
> **Comment 1.** The difference between the paper and previous work mainly relies on the (slightly generalized) assumptions. Hence, the contributions may not be groundbreaking.
>
> **Reply.** Thanks for this comment. We will add a table to compare our results with existing works for AdaGrad, including assumptions, convergence rate and the parameter setting to elaborate our contributions. In addition, we will add more details on proof challenges and our proof novelty to the main body, particularly using a new proxy step-size and temporary bounds to control unbounded function value gaps and unbound smooth parameters.
>
> ---
> Thanks a lot again for your positive comments. If you have any further questions, please feel free to ask.
>
> Best regards.
>
> Authors

---

### Official Review · Reviewer_XzZb · 2024-03-23

**Q2-1 Originality-Novelty:** 3
**Q2-2 Correctness-Technical Quality:** 3
**Q2-5 Clarity Of Writing:** 3

**Q1 Summary And Contributions:**

The ssubmitted draft discusses the convergence of AdaGrad with momentum on non-convex smooth optimization problems under a general noise model. The main contributions of the study include:

Demonstrating the probabilistic convergenc of AdaGrad with momentum on non-convex smooth optimization with a general noise assumption.
Deriving convergence results that are adaptive to noise levels and optimal in terms of convergence rate.
Providing convergence bounds for AdaGrad with momentum, considering noise parameters and the necessity of tuning step-sizes.
The analysis in the study relies on the descant lemma, decomposition techniques, and novel estimations related to proxy step-sizes. The convergence results are shown to be controlled by polynomial functions of log(T), indicating the optimization process's efficiency.

**Q2-3 Extent To Which Claims Are Supported By Evidence:**

3: Good: the main claims are supported by convincing evidence (in the form of adequate experimental evaluation, proofs, (pseudo-)code, references, assumptions).

**Q2-4 Reproducibility:**

3: Good: key resources (e.g. proofs, code, data) are available and key details (e.g. proofs, experimental setup) are sufficiently well-described for competent researchers to confidently reproduce the main results.

**Q3 Main Strengths:**

This paper on the convergence of AdaGrad with relaxed assumptions and has following demonstraated strenghts.

Novelty of Contribution : The study presents new insights into the convergence behavior of AdaGrad with momentum under a general noise model, expanding the understanding of optimization algorithms in non-convex smooth landscapes. The adaptive nature of the convergence results to noise levels and the optimal convergence rate showcases the effectiveness and efficiency of AdaGrad with momentum in handling varying noise conditions. The consideration of a general noise model that reflects real-world scenarios makes the findings of the study highly relevant for practical applications in various fields where optimization problems with noise are common.the paper contributes non-trivially to the advancement of optimization algorithms and can potentially influence future research directions in the field.

Rigorous analysis:  The analysis in the paper is based on rigorous mathematical frameworks, including the use of descent lemma, decomposition techniques, and novel estimations, ensuring the validity and robustness of the results.

Clear Presentation: The paper provides a clear and structured presentation of the problem setup, assumptions, methodology, and results, making it accessible to a wide audience of researchers and practitioners in the field of optimization. The paper was a pleasure to read.

**Q4 Main Weakness:**

I could only go through a subset of proofs, but I do not doubt the results.

The paper lacks any empirical evaluation. Even though it is a theory paper, it would have been nice to verify/validate teh assumptions and results. However, i do not think this is a deal breaker, since the contribution of the paper otherwise is solid.

The paper could and should be expanded with limitations and open future threads. A conclusion/discussion section at the end would be helpful, even if some of the results are required to be moved to the appendix.

**Q5 Detailed Comments To The Authors:**

Other than the above weaknesses, I do not have any furhter comments. I do not see any obvious typos either.

**Q9 Complying With Reviewing Instructions:**

Yes

---

> ### Author Rebuttal · Authors · 2024-04-07
>
> We really thank for your positive review and constructive suggestions.
>
> ---
> **Comment 1.** *The paper lacks any empirical evaluation. Even though it is a theory paper, it would have been nice to verify/validate the assumptions and results. However, i do not think this is a deal breaker, since the contribution of the paper otherwise is solid.*
>
> **Reply.** We thank for your suggestion and will add experiments in the revised version. In particular, (Khaled and Richtárik, 2023) has already done some experiments using SGD with the expected smoothness. We will follow their experiment setting and make some comparisons with their experimental results which could be reported in the appendix.
>
>
> ---
> **Comment 2.** *The paper could and should be expanded with limitations and open future threads. A conclusion/discussion section at the end would be helpful, even if some of the results are required to be moved to the appendix.*
>
> **Reply.** Thanks a lot for this valuable suggestion. Following your comment and Reviewer XmTt and JikW, we will add the Conclusion/Limitations Section in the revised version as follows.
>
> **Conclusion.** In this paper, we provide high probability convergence bounds for AdaGrad and its momentum variant under the non-convex smooth optimization. In particular, we consider a mild noise model incorporating affine variance noise and the expected smoothness. We rely on a new proxy step-size and some delicate estimations to derive the bound. Our findings reveal that without using problem-parameters, AdaGrad is powerful enough to converge to the stationary point with a rate of $\tilde{\mathcal{O}}(1/\sqrt{T})$, particularly accelerating to $\tilde{\mathcal{O}}(1/T)$ when specific noise parameters are sufficiently low. Furthermore, we extend our framework to the generalized smooth case which allows for unbounded smooth parameters while maintaining the same convergence rate. However, problem-parameters are necessary to tune step-sizes under the generalized smoothness.
>
> **Limitations.** Although AdaGrad plays an important role in the adaptive method field, several other adaptive methods are more preferred in some of real applications, including RMSProp, Adam and AdamW. Therefore, it is also pertinent to consider these algorithms under relaxed assumptions. In addition, we are still unknown whether the same result could be achieved under an expected version of Assumption (A3). We will thus focus on the above two points in the future works. Finally, as we study a new assumption over AdaGrad, it would be more convincing to provide more experimental results to support the theoretical results.
>
> ---
> Thanks a lot again for your positive comments. If you have any further questions, please feel free to ask.
>
> Best regards.
>
> Authors

---

### Official Review · Reviewer_XmTt · 2024-03-23

**Q2-1 Originality-Novelty:** 2
**Q2-2 Correctness-Technical Quality:** 3
**Q2-5 Clarity Of Writing:** 3

**Q1 Summary And Contributions:**

This paper focuses on analyzing the convergence of AdaGrad. It provides two relaxed assumptions on the distribution of the stochastic noise and the smoothness of the problem. The authors provide proof under the relaxed assumptions for AdaGrad with momentum, which matches the existing bounds and recovers the results for stricter assumptions. The author uses an induction with high probability to first bound the function value gap and then uses the gap to bound the update bias (variance) of AdaGrad, and finally obtain the convergence result.

**Q2-3 Extent To Which Claims Are Supported By Evidence:**

4: Excellent: all claims are supported by very convincing evidence (in the form of comprehensive experimental evaluation, rigorous mathematical proofs, detailed (pseudo-)code, precise references, well-motivated and realistic assumptions) and the authors deliver what they promise.

**Q2-4 Reproducibility:**

4: Excellent: key resources (e.g. proofs, code, data) are available and key details (e.g. proof sketches, experimental setup) are comprehensively described for competent researchers to confidently and easily reproduce the main results.

**Q3 Main Strengths:**

1. Extension to relaxed assumptions. The proof of AdaGrad (and its momentum variant) relies on relaxed assumptions of the noise and problem.

2. Extension to momentum case. The analysis for the momentum variant requires extra effort to bound the error term of the update bias.

**Q4 Main Weakness:**

1. Novelty: Although the analysis uses a relaxed assumption, the overall proof structure is still similar to the one used in [R1]. The relaxation of the assumption on the variance only impacts the coefficient before $\Delta_x$, where $\Delta_x$ has already appeared in the original proof in [R1].

[R1] Attia, A., & Koren, T. (2023, July). SGD with AdaGrad stepsizes: full adaptivity with high probability to unknown parameters, unbounded gradients and affine variance. In International Conference on Machine Learning (pp. 1147-1171). PMLR.

**Q5 Detailed Comments To The Authors:**

I would consider convergence under generalized smoothness as the major contribution (and novelty) of the framework, and section 4 as a special case. The change in the organization of the paper may be beneficial.

It would be great if the author could include other momentum methods in the analysis (e.g., Adam, AdamW).

A3, the relaxed affine variance noise is still a bit restrictive. An expectation version might improve the significance of the paper.

**Q9 Complying With Reviewing Instructions:**

Yes

---

> ### Author Rebuttal · Authors · 2024-04-07
>
> Thanks a lot for your effort to review our manuscript and constructive comments.
>
> ---
> **Comment 1.** *Novelty: Although the analysis uses a relaxed assumption, the overall proof structure is still similar to the one used in [R1].*
>
> **Reply.** We agree with you that some of our proof steps are motivated by [R1], which has also been commented in several places of the paper. In particular, we will move the sentence "Following [Li and Orabona, 2020, Attia and Koren, 2023], we will first present several important technical lemmas." from Appendix A to the main body of this paper, emphasizing the importance of [R1]. In addition, we are willing to summarize our key proof novelty in the following.
>
> **(1). New proxy step-size.**
>
> We introduce a new type of proxy step-size $a_s$ relying on the temporary gradient bound $\mathcal{G}_s$ as follows,
> $$
> a\_s = \sqrt{v\_{s-1}+(\mathcal{G}\_s 1\_d)^2}.
> $$
> The idea of using proxy step-size is inspired by several previous works including [R1]. We modify their types to handle the more relaxed assumption (A3). As a consequence, we provide new estimations to control the error brought by $a_s$, specifically in Lemma 5.4 and Lemma 5.5.
>
> **(2). Rough/delicate estimations for controlling  $\Delta_s^{(x)}$.**
>
> We agree with your comment that the central proof idea is to control $\Delta_s^{(x)}$. In addition, we provide some more new technical details related to $\Delta_s^{(x)}$.
>
> After estimating separated parts in (10), we derive inequalities in (19) and (22). We are inspired by the estimation for the summation $\sum_{s=1}^t \|g_s\|^2/b_s$ in e.g., (Kavis et al., 2022, Attia et al., 2023) and further obtain estimations for the following coordinate-wise summations,
> $$
> \underbrace{\sum\_{s=1}^t \left\\|\frac{g\_s}{b\_s} \right\\|^2}\_{(A)} \le \sum\_{i=1}^d \log \left(1+\frac{1}{\epsilon^2}\sum\_{s=1}^t g\_{s,i}^2 \right),  \quad
> \underbrace{\sum\_{s=1}^t \\|m\_s\\|^2}\_{(B)} \le \frac{\eta^2}{(1-\beta)^2}\sum\_{i=1}^d \log \left(1+\frac{1}{\epsilon^2}\sum\_{s=1}^t g\_{s,i}^2 \right).
> $$
> If we use (A3) to control $g_{s,i}^2$, there remains $\Delta_s^{(x)}$ inside the logarithm term. Since in the next step we will use an induction argument to bound $\Delta_s^{(x)}$ , the $\log \left(\sum_{s=1}^t  \Delta_s^{(x)}\right)$ will bring more calculation complexity. We handle this by providing a rough estimation for $\Delta_s^{(x)}$ in Lemma 5.1 and Lemma 5.2 with $\mathcal{O}({\rm poly} (T))$ order. Therefore, the summations (A) and (B) are now controlled by $\mathcal{O}({\rm poly} (\log T))$ order.
>
> Next, it's necessary to further lower bound $\Delta_{s}^{(y)}$ by $\Delta_s^{(x)}$ in the LHS of (22), which is provided in Lemma 5.7 through the descent lemma. Moreover, the additional gap $\|m_s\|$ is controlled by $\mathcal{O}({\rm poly} (\log T))$ in Lemma 5.8.
>
> **(3). Better dependency to $(1-\beta)^{-1}$.**
>
> As pointed out by (Défossez et al., 2020), reducing the dependency of $(1-\beta)^{-1}$ is significant. Hence, we aim to achieve the best known dependency $\mathcal{O}((1-\beta)^{-1})$. A key technical step is to bound $\|m_s\|^2$ in Lemma 5.8 with $\mathcal{O}((1-\beta)^{-1})$ instead of $\mathcal{O}((1-\beta)^{-2})$ in Lemma 5.2. Specifically, if we use Lemma 5.8 and Lemma 5.2 respectively, the convergence bound satisfies that
> $$
> \mathcal{O}\left(\left(\frac{1-\beta}{\eta} + \frac{\eta}{(1-\beta)^2} \right)^2\right), \quad \text{or} \quad  \mathcal{O}\left( \left(\frac{1-\beta}{\eta} + \frac{\eta}{(1-\beta)^3} \right)^2\right),
> $$
> which would lead to the optimal dependency $\mathcal{O}((1-\beta)^{-1})$ and  $\mathcal{O}((1-\beta)^{-2})$ respectively. However, Lemma 5.2 is also important as the estimation for $\|m_s\|^2$ in Lemma 5.8 relies on these rough estimations in Lemma 5.2.
>
> **(4). Controlling unbounded smooth parameter under generalized smoothness.**
>
> To handle this challenge, we use $\mathcal{L}_s =2 L_0 +2 L_1\left(4L_1 \Delta_s^{(x)}+\sqrt{4L_0\Delta_s^{(x)}}\right)$ to upper bound the generalized smooth parameter. The key insight is that when we use a similar induction argument as the smooth case, supposing $\Delta_s^{(x)} \le \Delta, \forall s \le t$, it will consequently lead to $\mathcal{L}_s \le \mathcal{L},\forall s \le t$. Moreover, since $\mathcal{L}$ is given in advance, we could let $\eta \sim \mathcal{O}(1/\mathcal{L})$ to control the unbounded smooth parameter, at the expense of using problem-parameters to tune $\eta$. It's worthy noting that the prior information is necessary, which is also indicated by the counter example given in (Wang et al., 2023).
>
> We also quite thank for the positive review for the generalized smooth part in your comment.
>
> ---
> *To be continued...*

---

### Meta-Review · Area_Chair_NdPL · 2024-04-17

The paper presents an insightful analysis of AdaGrad and its momentum variant
under relaxed assumptions, contributing to the understanding of how these
popular optimization algorithms perform under more practical, less ideal
conditions. The authors have provided a rigorous mathematical framework to
support their claims and have demonstrated the correctness of their approach
through comprehensive proofs and empirical evidence.



Pros
+ The paper provides a novel analysis of AdaGrad with momentum under relaxed assumptions regarding the stochastic noise and smoothness of the problem.
+ Offers comprehensive and rigorous mathematical proofs supporting the claims about convergence. Includes analysis for both AdaGrad and its momentum variant, demonstrating the utility under broader conditions.
+ Excellent reproducibility with detailed resources like proofs, code, and data made available.
+ Despite some areas needing improvement, the paper is generally well-organized.

Cons
- The relaxed assumptions, while useful, do not dramatically change the fundamental analysis of AdaGrad, and the proof structure closely resembles existing works.
- The practical impact and broad applicability of the findings are not strongly highlighted, possibly limiting the interest of a broader audience.
- Some parts of the paper could benefit from clearer explanations to enhance understanding, especially for readers not already familiar with the topic.

I recommend acceptance of this paper. It would be advisable for the authors to
consider the reviewers' feedback to refine the presentation and further clarify
the significance and novelty of their work in relation to existing literature.
With these enhancements, the paper could appeal to a wider segment of the
community and increase its impact. Still, these improvements can be done when
preparing camera ready paper.